# Programmed disassembly of a microtubule-based membrane protrusion network coordinates 3D epithelial morphogenesis in *Drosophila*

Ngan Vi Tran[1], Martti P Montanari[2], Jinghua Gui [2], Dmitri Lubenets[1], Léa Louise Fischbach [2], Hanna Antson[1], Yunxian Huang[2], Erich Brutus[1], Yasushi Okada [3,4], Yukitaka Ishimoto[5], Tambet Tõnissoo[1] & Osamu Shimmi [1,2 ✉]

## Abstract

**Comprehensive analysis of cellular dynamics during the process of morphogenesis is fundamental to understanding the principles of animal development. Despite recent advancements in light microscopy, how successive cell shape changes lead to complex three-dimensional tissue morphogenesis is still largely unresolved. Using in vivo live imaging of *Drosophila* wing development, we have studied unique cellular structures comprising a microtubule-based membrane protrusion network. This network, which we name here the Interplanar Amida Network (IPAN), links the two wing epithelium leaflets. Initially, the IPAN sustains cell–cell contacts between the two layers of the wing epithelium through basal protrusions. Subsequent disassembly of the IPAN involves loss of these contacts, with concomitant degeneration of aligned microtubules. These processes are both autonomously and non-autonomously required for mitosis, leading to coordinated tissue proliferation between two wing epithelia. Our findings further reveal that a microtubule organization switch from non-centrosomal to centrosomal microtubule-organizing centers (MTOCs) at the G2/M transition leads to disassembly of non-centrosomal microtubule-derived IPAN protrusions. These findings exemplify how cell shape change-mediated loss of inter-tissue contacts results in 3D tissue morphogenesis.**

**Keywords** Epithelial Morphogenesis; Cellular Protrusion; Three-Dimensional Morphogenesis; Microtubule Dynamics; Non-Centrosomal Microtubule Organizing Center
**Subject Categories** Cell Adhesion, Polarity & Cytoskeleton; Development

## Introduction

Epithelial cells are one of the basic units of organogenesis in animal development. To generate proper tissue and complex 3D organs, cell shape must be adapted to sustain overall tissue morphogenesis (Gómez-Gálvez et al, 2021). Since cell shapes change dynamically, it is crucial to employ a system that enables observation of real time cell shape changes during morphogenesis (Hannezo and Heisenberg, 2019). Various cell shapes include formation of membrane protrusions in divergent developing tissues (Davidson et al, 2008; Demontis and Dahmann, 2007; Ramirez-Weber and Kornberg, 1999; Sagar et al, 2015; Sato et al, 2017). Although our understanding how these protrusions contribute to morphogenesis is still limited, recent studies have provided new insights. For example, tubular membrane protrusions have been shown to be involved in signal transduction in various contexts: actin-based filopodia-like structures called cytonemes serve to sustain long-range paracrine signaling in various tissues and species (Kornberg, 2017); microtubule (MT)-based nanotubes and cytosensors transduce a short-range BMP signal in *Drosophila* germline stem cells (Inaba et al, 2015; Wilcockson and Ashe, 2019); and MT-based protrusions termed primary cilia are known to be involved in signaling pathways in vertebrate cells (Corbit et al, 2005; Huangfu and Anderson, 2005).

Furthermore, various forms of membrane protrusions have been described in tissue culture cells (Rustom et al, 2004). Tunneling nanotubes (TNTs) have been proposed to form an open-ended network composed of microfilament (MF)-based extensions between cells to sustain a cellular network and to help move vesicles and organelles (Cordero Cervantes and Zurzolo, 2021; Rustom et al, 2004). TNTs and other types of membrane protrusions play important roles in pathological conditions as well, serving as a platform to transfer pathogens such as viruses and aggregated proteins (Hashimoto et al, 2016; Osswald et al, 2015; Scheiblich et al, 2021). However, the physiological roles of such structures are poorly understood due to limited accessibility of in vivo models.

[1]Institute of Molecular and Cell Biology, University of Tartu, 51010 Tartu, Estonia. [2]Institute of Biotechnology, University of Helsinki, 00014 Helsinki, Finland. [3]Center for Biosystems Dynamics Research, RIKEN, Osaka, Japan. [4]Departments of Cell Biology and Physics, University of Tokyo, Tokyo, Japan. [5]Department of Machine Intelligence and Systems Engineering, Akita Prefectural University, Akita 015-0055, Japan. ✉E-mail: osamu.shimmi@helsinki.fi

MTs are one of the basic elements of the cytoskeleton. Recent studies showed that MT nucleation is mediated by different types of MT organizing centers (MTOCs) in a context-dependent manner (Muroyama and Lechler, 2017; Wu and Akhmanova, 2017). When cells undergo mitosis, MTs are typically nucleated by centrosomal MTOCs (cMTOCs) (Wu and Akhmanova, 2017). On the other hand, within differentiated cells, MT nucleation is often organized by non-centrosomal MTOCs (ncMTOCs) (Muroyama and Lechler, 2017). A study in *Drosophila* salivary gland showed that MTs in mitotic cells are gradually reorganized from cMTOCs to ncMTOCs during morphogenesis, and these processes are coupled to a dynamic MF and Myosin II network (Booth et al, 2014; Röper, 2020). Therefore, dynamic cell shape changes involve dynamic changes in both MTs and MFs.

Wing development in *Drosophila* is a classical model for studying genetic control of tissue morphogenesis. Previous studies reveal that a number of conserved signaling pathways act in the developing wing to control its size, shape and patterning, and a number of genetic tools exist to enable further studies (Blair, 2007). The larval wing imaginal disc, a single-layer epithelium, is widely used as a model to study the integration of diverse regulatory cues for tissue growth and pattern formation (Tripathi and Irvine, 2022). During metamorphosis, the single-layered wing imaginal disc becomes a two-layered pupal primordial wing comprising dorsal and ventral epithelium (Fig. 1A). During the first 24 h after puparium formation (APF), wing development is divided into three phases. In the first (0–10 h APF, first apposition), the wing disc becomes a rudimentary two-layered wing. In the second (10–20 h APF, inflation), the two epithelia separate, and appose again in the third phase (at 20 h APF, second apposition) (Fristrom et al, 1993; Gui et al, 2019; Montanari et al, 2022). In the pupal wing, cell proliferation mainly occurs during the inflation stage (Etournay et al, 2016; Milan et al, 1996). Dynamics of pupal wing development have been described in pioneering studies by Conrad Waddington (Waddington, 1940). Recent reports further reveal that *Drosophila* pupal wing serves as an attractive model to address the mechanisms behind planar cell polarity (PCP) and epithelial cell packing by combining in vivo live image analysis with *Drosophila* genetics (Aigouy et al, 2010; Etournay et al, 2015).

The developing *Drosophila* pupal wing comprises two epithelia essentially identical to each other in size, shape and patterning. Previous studies have suggested the existence of communication between dorsal and ventral cells (Garcia-Bellido, 1977), and recent studies further reveal the importance of communication between the two epithelia for wing growth and patterning (Gui et al, 2019). However, how growth of the two epithelia is coordinated during the proliferation/inflation stage in spite of their physical separation remains poorly understood. Previous studies have provided clues to address this question. Unique cellular structures called transalar cytoskeletal arrays (TCAs) are composed of both MTs and MFs that form physical links between the dorsal and ventral epithelia (Fristrom et al, 1993). While we were preparing this work, it has been reported that such structures are observed to form just before the inflation stage at around 8 h APF in in vivo live imaging, and TCAs appear to be connected between the two epithelial layers via a basal integrin-laminin complex, revealing that unique cellular mechanisms may exist to sustain 3D organ formation during pupal wing development (Sun et al, 2021). Although it has been argued that TCAs serve as a means to support apposition of the dorsal and ventral epithelia to sustain proper adult wing structure, how they are involved in wing growth remains to be addressed.

Here, using *Drosophila* pupal wing as a model, we address unique cellular mechanisms supporting 3D epithelial morphogenesis. By establishing a non-invasive live imaging protocol to observe pupal wing development, we characterize a highly dynamic intercellular network of MT- and MF-based membrane protrusions between the two epithelia. Although a part of the structure has been previously described as TCAs (Fristrom et al, 1993; Sun et al, 2021), our findings include both vertical MT protrusions and lateral filopodia-like structures resembling Amida (amida-kuji (ghost leg) is a Japanese term describing a ladder-like network). We further observed that the vertical protrusions are actively networked by filopodia-like lateral structures, which lead to the formation of bundled protrusions in a gradual process. Thus, we term the structure the Interplanar Amida Network (IPAN) to more effectively describe its structure and functions. Our data reveal that disassembly of the IPAN leads to loss of cell–cell contact between the two epithelia, resulting in coordinated mitosis. Employing quantitative functional analysis based on in vivo live imaging, we demonstrated that MT and MF co-factors are required for the process of programmed disassembly of the IPAN and subsequent coordinated tissue proliferation. Cell proliferation is then regulated by G2/M transition executors in a cell autonomous manner. Finally, our data show that MT-based protrusions of the IPAN are nucleated by ncMTOCs. As the IPAN disassembles, MTs degenerate, and then reform as centrosome-based mitotic spindle MTs in proliferating cells. Taken together, our data reveal that the unique cellular structure of the IPAN provides a novel cellular mechanism in 3D morphogenesis.

## Results

### The *Drosophila* pupal wing forms a cellular network comprising membrane protrusions between dorsal and ventral epithelia

Although the *Drosophila* pupal wing serves as a unique model for a 3D organ formation (Gui et al, 2019), pupal wing during the inflation stage had proven refractory to recovery, fixation, and subsequent investigation of structural details. To overcome these limitations, we employed a live imaging approach that allows us to observe dynamics of cellular structures. At the end of the prepupal stage, head eversion takes place (Fig. 1A; Waddington, 1940). Concurrently, the position of the developing pupal wing shifts mediolaterally from the anterior. When head eversion is complete, the pupal wing position stabilizes at ~13 h APF (25 °C). A small window in the pupal case that exposes the dorsal surface of the pupal wing and the hinge region can then be excised. A droplet of halocarbon oil to prevent dehydration is applied to the exposed pupal wing, and the pupa is mounted on a cover glass for live imaging (Fig. 1B; Classen et al, 2008). Importantly, this protocol allows pupal flies to continue essentially normal development: most live-imaged pupae mature into adults with anatomically normal wings, indicating our experimental procedure does not significantly compromise normal physiological conditions.

When pupal wings expressing GFP-tagged αTubulin were imaged live by confocal microscopy at ~13–14 h APF, unique

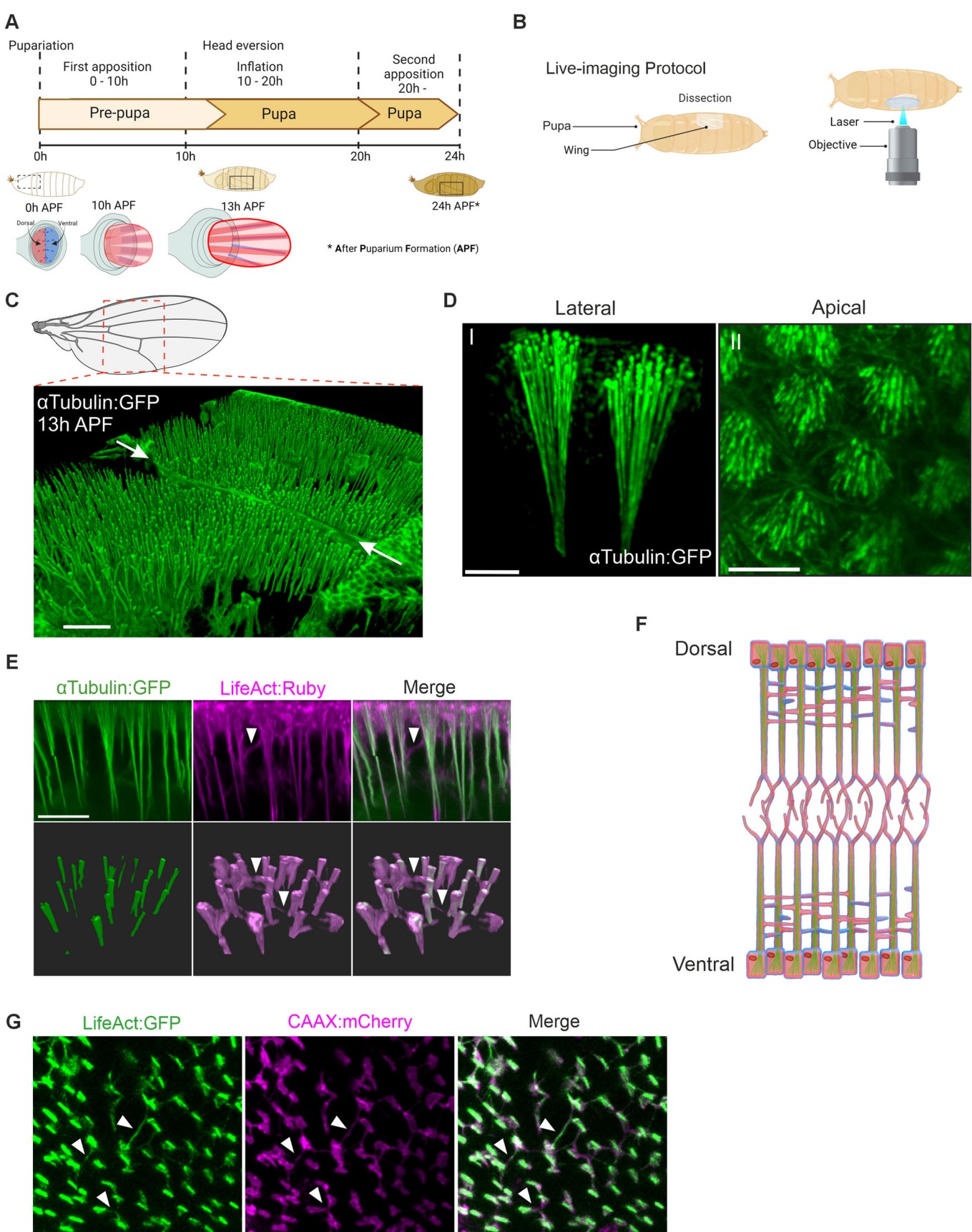

◀  **Figure 1.   The Interplanar Amida Network (IPAN) comprises microtubule vertical projections interlinked by lateral filopodia-like structures.**

(A) Schematic depicting the developmental time frame in which the pupal wing is formed during the pre-pupal stage (wing indicated by rectangle). The IPAN is formed prior to head eversion, and is further elaborated soon after head eversion as the wing elongates proximodistally. Schematics of each developmental stage of wing progenitors are shown below (dorsal in magenta and ventral in blue). (B) Graphic summary of the live imaging protocol. A window is cut into the pupal case of a live pupa in the proximal wing region after head eversion. Live imaging by confocal microscopy is performed within the window as the wing develops. (C) 3D view of MT protrusions visualized by αTubulin:GFP at 13 h APF. The dotted red box in the schematic of the adult fly wing shows the approximate location of the region targeted in the pupal wing during live imaging. Apical surface of the dorsal epithelium is towards the top of the page. White arrows denote the wing trachea. MTs protrude towards the bottom of the page into the interepithelial space. (D) High resolution images of αTubulin:GFP at 13 h APF. Panel I: MT protrusions comprise individual MTs that emanate from the apical surface of the cell and bundle as they extend basally. Panel II: The zoom-out provides an apical view of MTs that bundle further basally (out of the plane of view). (E) The IPAN comprises apicobasally oriented MT protrusions (green, αTubulin:GFP) connected laterally by MFs (magenta, LifeAct:Ruby). Upper panels provide an optical cross-sectional, and lower, an oblique view (apical towards the top of the page). White arrowheads point at lateral MFs. (F) Schematic of the IPAN at ~13–14 APF (25 °C). Green: MT protrusions. Red: F-actin. Blue: plasma membrane. (G) IPAN in the interepithelial plane. MFs (LifeAct:GFP, green) and cell membrane (CAAX:mCherry, magenta) that both envelop and connect the MT protrusions are shown. White arrowheads point at lateral filopodia-like structures connecting MT protrusions. Scale bars: 40 μm (C), 5 μm (D), 10 μm (upper panel in E, G). See also Figure EV1 and Movie EV1.

intracellular structures were visualized: MT protrusions are observed that originate in the apical region of the cell and extend basally (Fig. 1C).

To understand in detail how MT projections are formed, we employed high resolution live imaging. When pupal wings expressing αTubulin:GFP and LifeAct:Ruby, which visualizes actin filaments, were imaged, we found that MT foci are distributed medioapically around 13–14 h APF (Fig. 1D, Movie EV1). On average, ~30 such apical MT foci are found per cell (Fig. EV1A,B). The MT protrusions are surrounded by actin filaments (Fig. 1E,F, Movie EV1). In addition, lateral extensions contain actin filaments, but not MTs. Therefore, the MT-based membrane protrusions are formed vertically, and a lateral network between vertical protrusions is formed by filopodia-like structures (Fig. 1E,F). This was further confirmed with membrane-bound CAAX:mCherry, which shows a similar localization to LifeAct (Figs. 1G and EV1C). Whereas the vertical MT projections are prominent and robust, the lateral MF extensions are thin and threadlike. This 3D meshwork structure is formed between the two epithelial sheets (Fig. 1F, Movie EV1). We further confirmed that MT projections assemble away from the nucleus throughout the cell body (Fig. EV1D). To effectively describe its structure and functions, we term the structure the Interplanar Amida Network (IPAN) henceforward.

## The IPAN is dynamic and transient

Time-lapse imaging of the IPAN by employing membrane-bound CAAX:mCherry reveals that the membrane protrusions are not stable, but instead transient, with three unique features. First, a subset of protrusions is disassembled in a time-dependent manner. Second, the remaining protrusions initiate bundling, first locally with neighboring protrusions, then subsequently either exiting the bundles for disassembly, or extending proximodistally in bundle fronts throughout the wing, eventually forming an interepithelial cytoskeletal basement that mediates apposition of the two epithelia (described previously as the cytoskeletal sheet, Fig. 2A, Movie EV2; Fristrom et al, 1993). Third, membrane protrusions are found in the regions of the future wing blade and hinge. Although hinge contraction begins before the second apposition stage (Etournay et al, 2016), the IPAN dynamics appear to be similarly coordinated in the future wing blade and hinge until 18 h APF.

To comprehend how vertical MT protrusions are regulated during these processes, we conducted time-lapse imaging using

wings that express αTubulin:GFP and CAAX:mCherry. Our observations reveal that lateral filopodia-like structures both connect and exhibit an active movement back and forth between the vertical protrusions (Fig. 2B, Movie EV3). Occasionally, a relatively small number of vertical protrusions disappears. Interestingly, during the early phase, vertical protrusions remain in the same position. When MT projections become thinner, vertical protrusions appear to be flexible and tend to form a bundle (Fig. 2C, Movie EV3). Initial bundles comprise several vertical protrusions, which we term "the primary bundle". These results indicate that MT projections can have two distinct fates: (1) Either MT projections become disassembled or (2) MT projections become thinner, but integrate into a bundle, subsequently forming a higher-order bundles.

Time-lapse imaging of the αTubulin:GFP-labeled IPAN at the tissue level further confirms that individual MT projections form a primary bundle, followed by higher-order bundle formation, which eventually extends proximodistally across the tissue. We noticed that some of the primary bundles located in the flanking region of future wing veins become disassembled (Fig. 2D, Movies EV4). Conversely, higher-order bundle structures accumulate in the middle of future intervein regions prior to the second apposition stage. Taken together, these results indicate that the IPAN dynamics are regulated by coordinated mechanisms between vertical MT projections and lateral filopodia-like structures.

## The IPAN is composed of membrane protrusions emanating from both dorsal and ventral cells to sustain cell–cell contacts

We next investigated how the two epithelia form the IPAN. Although previous studies demonstrated that a basal matrix is required for maintaining protrusions (Sun et al, 2021), it remains to be addressed how dorsal and ventral protrusions are formed to interact with each other. To distinguish the cell membranes derived from dorsal and ventral epithelial sheets, we generated a two-color system in which membrane-bound RFP and -GFP were expressed in dorsal and ventral cells, respectively (Fig. 3A). Using this system, we first unveiled the structures of dorsal and ventral protrusions with high resolution imaging. Protrusions are seen as a defined intracellular extension of the basal domain. Interestingly, the basal compartment of the cell body displays a single protrusion per cell, but these protrusions form several branches prior to reaching the

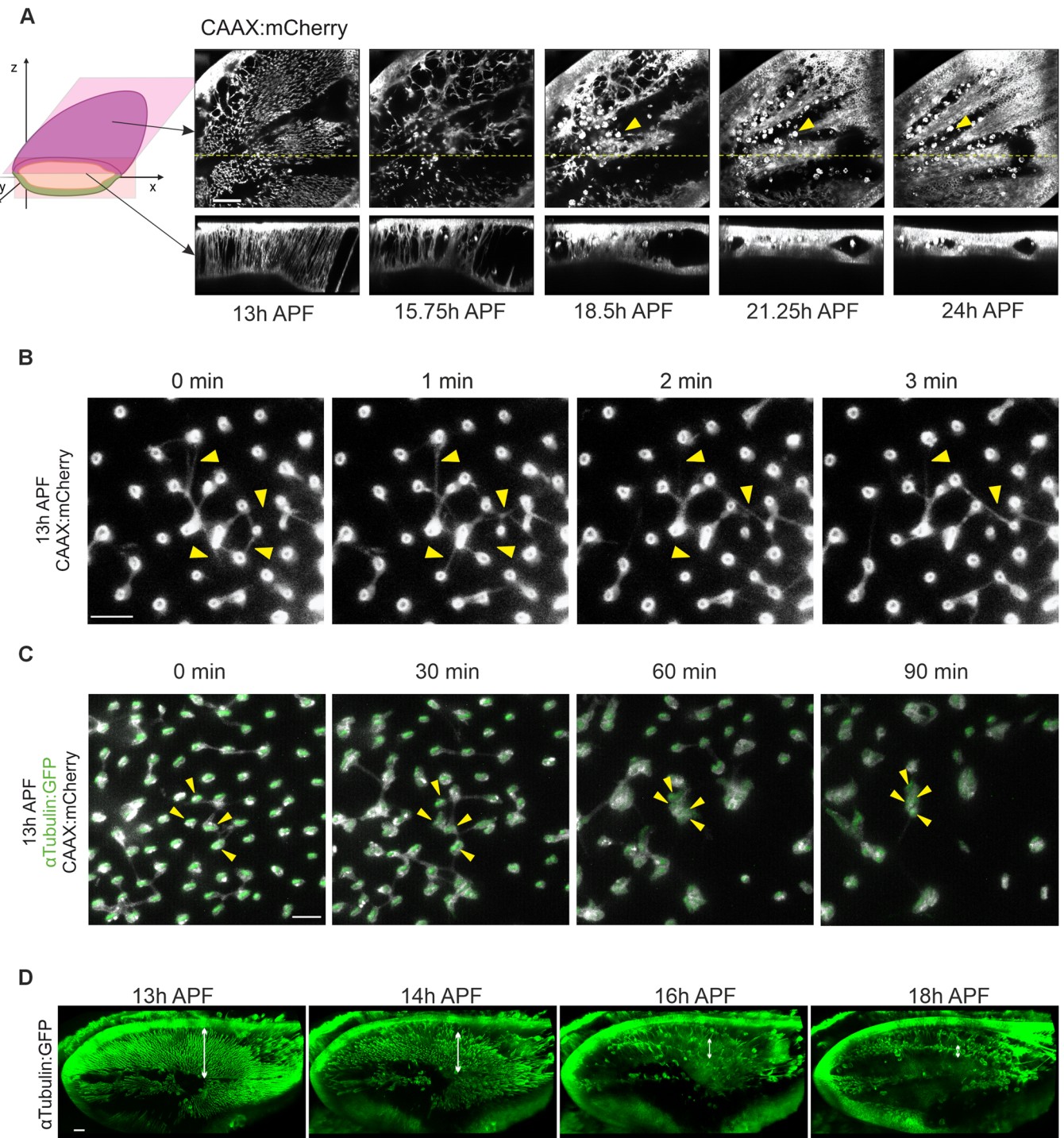

**Figure 2. Structures of the IPAN are dynamic and transient.**

(A) Time-lapse images of CAAX:mCherry (white) of the pupal wing between 13 and 24 h APF. Upper panels show interepithelial transverse (*XY*) views of the IPAN at the different time points. Dotted yellow lines are the approximate regions of the optical cross-section views in the corresponding lower panels. Bright amorphous objects (yellow arrowheads) are macrophage-like hemocytes. A corresponding time-lapse movie is shown in Movie EV2. Schematic at left shows approximate positions at which the live images were generated. (B) Time-lapse images of an apical view of CAAX.mCherry starting at 13 h APF. A corresponding time-lapse movie is shown in Movie EV3. Arrowheads point at lateral filopodia-like structures connecting vertical protrusions. (C) Time-lapse images of an apical view of CAAX.mCherry (white) and αTubulin:GFP (green) starting at 13 h APF. A corresponding time-lapse movie is shown in Movie EV3. Arrowheads point at bundling of vertical protrusions. (D) Time-lapse images of αTubulin:GFP in pupal wing starting at 13 h APF. Apical surface of the dorsal epithelium is towards the top of each panel. A corresponding time-lapse movie is shown in Movie EV4. Note that higher-order bundle structures are observed in the middle of future intervein regions marked by double-ended arrows. Scale bars: 50 μm (A), 5 μm (B, C), 30 μm (D).

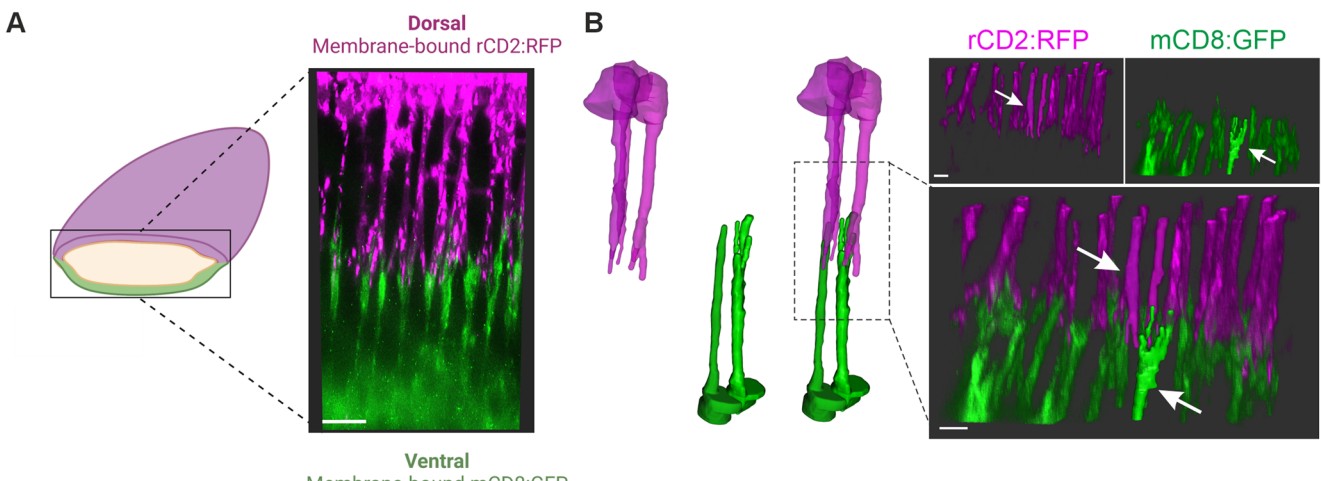

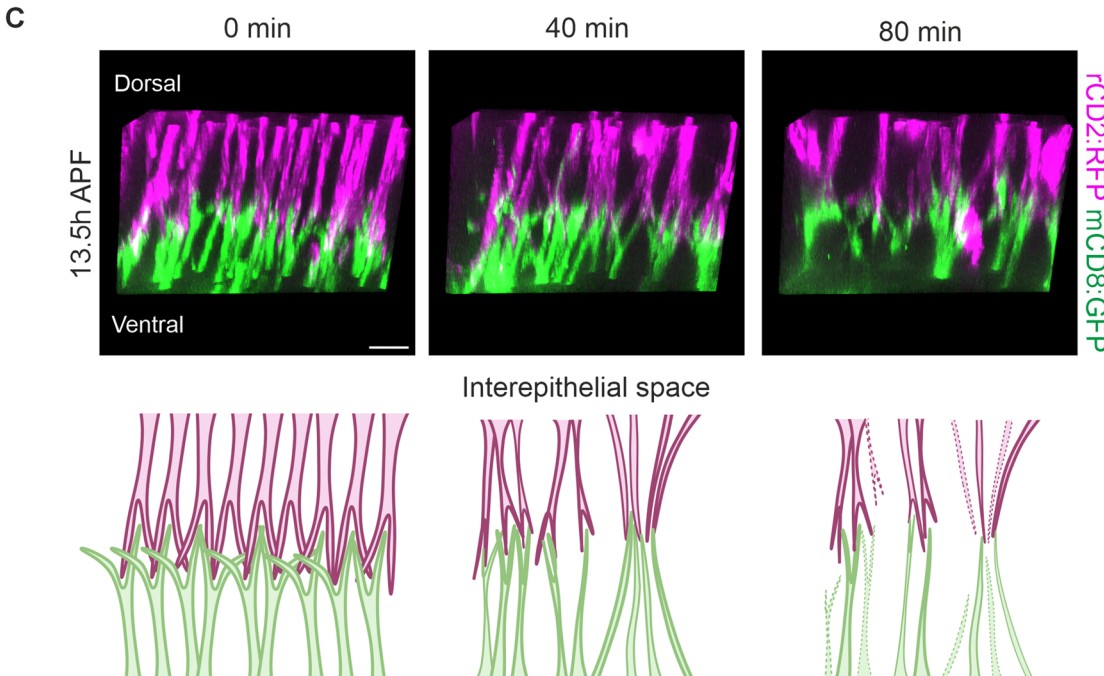

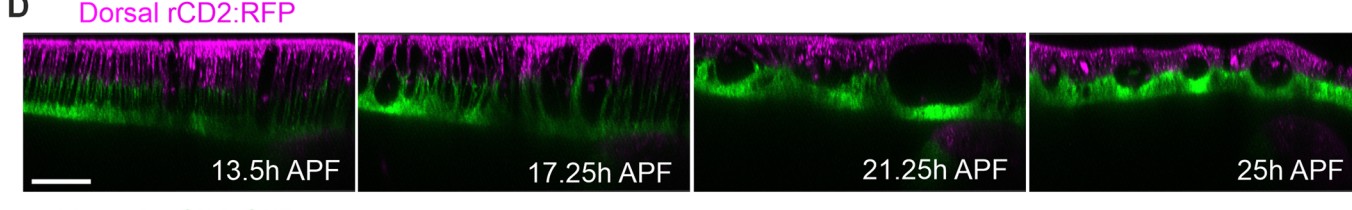

center space between the two epithelial sheets. These branches in turn form contacts with the branches of the protrusions from the opposite epithelial sheet (Fig. 3A–C, Movie EV5). Branched basal contacts from a single cell appear to interact with branches derived from multiple cells between the epithelia, and thus interepithelial

cell–cell contacts via protrusions are not mediated by individual apposed dorsal and ventral cell pairs, but instead via multiple cells between the two layers.

To elucidate the dynamics of the IPAN, we employed time-lapse imaging. Our observations reveal that membrane protrusions are

**Figure 3.  MT protrusions of the IPAN emanate from dorsal and ventral epithelia to sustain cell–cell contacts.**

(A) The IPAN emanates from dorsal and ventral epithelia and branches basally to interact with protrusions from the apposed layer in the interepithelial space. Membrane-bound rCD2:RFP (magenta) is expressed dorsally and mCD8:GFP ventrally. (B) 3D rendering of representative differentially colored dorsal and ventral MT protrusions (left panel). Detailed oblique view of differentially colored dorsal and ventral protrusion branches interdigitating in the interepithelial space. White arrows indicate 3D-rendered protrusions at left (box with dotted lines). (See also Movie EV5.) (C) Upper panel: Time-lapse images of rCD2:RFP (dorsal) and mCD8:GFP (ventral) of the pupal wing starting at 13.5 h APF. Oblique view of interdigitating protrusions at the different time points. Lower panel: Schematics of cell–cell contacts formed by dorsal and ventral protrusions in the interepithelial space. (D) Time-lapse images of rCD2:RFP (dorsal) and mCD8:GFP (ventral) of the pupal wing between 13.5 and 25 h APF. Optical cross-sectional view at the different time points. (See also Movie EV5.) Scale bars: 10 μm (A, C), 5 μm (B), 50 μm (D).

generated from both dorsal and ventral cells, which meet in the interepithelial space (Fig. 3C,D, Movie EV5). Protrusions from the two epithelia interact within a ~10 μm-thick layer between the dorsal and ventral epithelia around 13 h APF (Fig. 3B), suggesting that cell–cell contacts between the two epithelia are maintained through membrane-membrane interactions. Moreover, bundling and disassembly of the IPAN take place in both dorsal and ventral layers in a coordinated manner (Fig. 3C,D, Movie EV5). Therefore, dorsal and ventral protrusions form a complex network through cell–cell contacts, which may play a key role in sustaining the dynamic IPAN structure.

## Disassembly of the IPAN involves MT reorganization and mitosis in pupal wing epithelial cells

When the IPAN is disassembled, the spatial distribution of MTs changes significantly. A large part of the intracellular αTubulin:GFP pool appears to be utilized for MT projections at 13–14 h APF (Fig. 1D). Then, as the MT protrusions regress, the distribution of apical lateral MTs becomes more apparent, the number of apical MT foci gradually decreases, and the mitotic spindle starts forming after the loss of vertical MT projections. To clarify the change of the cellular structures over time, we aimed to understand how sequential processes from the disassembly of MT projections take place by employing a basal view of time-lapse images. In addition to αTubulin:GFP, we used the pericentriolar material (PCM) marker Centrosomin (Cnn) to track mitoses in live imaging of the pupal wing (Basto et al, 2008). When αTubulin:GFP and Cnn:RFP are co-expressed, mitotic cells are clearly visualized by spindle formation and appearance of Cnn foci before cell division (Fig. 4A). Cnn signal is observed as a single focus prior to spindle formation, but two foci form before cell division. After cell division, Cnn foci degenerate and can no longer be visualized in our live imaging. Time-lapse images reveal that MT vertical projections degenerate prior to forming mitotic spindles (Fig. 4A, Movie EV6). To precisely monitor the cell cycle, we further conducted time-lapse imaging using the S/G2/M-green system (Nakajima et al, 2011). Our observations reveal that the majority of cells remain in the S/G2 phase during the early inflation stage as reported previously (Fig. EV2A; Milan et al, 1996). Subsequently, cells in M phase begin to appear, which aligns with the data obtained from Cnn:RFP and αTubulin:GFP (Fig. EV2A). We therefore conclude that Cnn:RFP foci together with αTubulin:GFP-labeled mitotic spindles provide a suitable readout for cell division.

Our observations are summarized as follows: The majority of αTubulin is utilized for membrane protrusions; then vertical MT projections partially degenerate; vertical MT projections mostly

degenerate; and αTubulin is utilized to form the mitotic spindle, and mitotic cell rounding occurs (Fig. 4A,B; Gibson et al, 2006).

We next addressed how MT projection dynamics affect cell–cell contacts between the two- epithelia. Our observations include that the loss of cell–cell contact takes place both prior to and after primary bundle formation (Fig. 4C–E). It has been proposed that cell–cell contact is mediated basal integrin-laminin complex (Sun et al, 2021), we therefore hypothesize that disassembly of MT projections involves concomitant degeneration of extracellular matrix (ECM), leading to the loss of cell–cell contact (Fig. 4F).

## Coordinated mitosis takes place between dorsal and ventral epithelia

Our observations that the sequential process from MT protrusions to mitosis (Fig. 4A), and the disassembly of MT projections involves the loss of cell–cell contact (Fig. 4C–E), enable us to investigate how changes in MT structures affect mitosis in the two epithelia. The ventral epithelium bulges in the distal region of the pupal wing, resulting in an interepithelial distance of ~150 μm, a distance that precludes obtaining images of sufficient quality. Therefore, we focused on the region covering proximal wing blade and hinge to quantify mitoses and MT projection dynamics in both dorsal and ventral epithelia, where the interepithelial distance between dorsal and ventral epithelia is smaller (60–80 μm), and the epithelia can be imaged at sufficient resolution (Fig. EV2B). Under these conditions, live imaging of the two epithelia can be straightforwardly carried out by confocal microscopy. To quantify the dynamics of MT projections and mitoses, 48 time-lapse images are captured every five minutes for four hours from 10.5 to 14.5 h APF at 29 °C, which corresponds approximately to 13.5–18 h APF at 25 °C (Ashburner et al, 2004). For the sake of simplicity, and unless otherwise mentioned, we henceforward refer to time after puparium formation at 29 °C. By focusing on the area proximal to the future anterior crossvein, where cells have smaller apical domains than elsewhere (Fig. EV2B), we can consistently capture the same region of interest (ROI) in the wing for quantification from one wing to the next. When in vivo 5D images (*xyz* spatial dimensions, time, and multiple excitation wavelength dimensions) are taken during a 4-h period, most cells contain MT projections at 10.5 h APF, and approximately 300–400 cells are found in each cell layer within the ROI (Figs. 5A and EV2B).

To understand how the MT projection dynamics are coupled with mitoses in the two epithelia, we analyzed the structural changes of MT projections. Since we have observed that the disassembly of MT projections proceeds concomitantly with mitosis (Fig. 4A, Movies EV6), we investigated a 3D view of MT projections during a 4-hour period. At the beginning of time-lapse

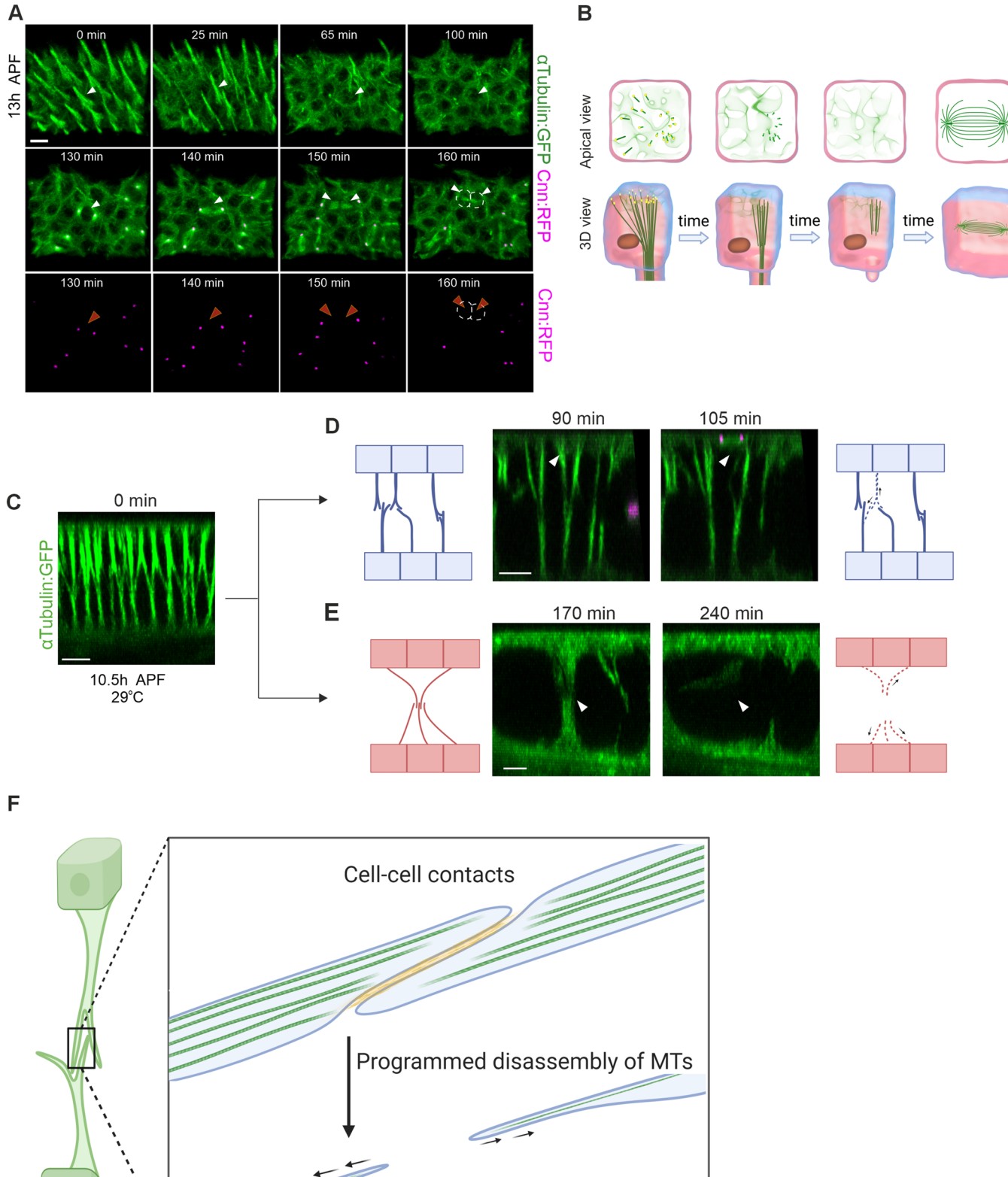

**Figure 4. Disassembly of MT protrusion and subsequent mitosis sequentially occur in the wing epithelia.**

(A) Basal view of αTubulin:GFP (green) and Cnn:RFP (magenta) of the dorsal epithelium starting at 13 h APF (25 °C) (top and middle images). Basal view of Cnn:RFP (magenta) of the dorsal epithelium between 130–160 min after 13 h APF (bottom images).The arrowheads point to cells tracked in time-lapse, and dashed lines mark two cells after division. (See also Movie EV6.) (B) Schematic of MT structural changes in which MT protrusions regress, with subsequent mitosis. First panel: individual MTs have clear apical foci (top row, apical view), and join to form MT projections further basally (bottom row, lateral view). Second panel: Fewer individual MT foci are found apically, and basal MT projections are less focused. Third panel: MTs no longer form apical foci, and the MT projections have disassembled. Fourth panel: A MT spindle and mitotic cell rounding are observed as the cell undergoes mitosis. (C–E) Structural changes of MTs involve loss of cell–cell contacts. Most cells contain basal protrusions to sustain cell–cell contacts between dorsal and ventral epithelia (C). Loss of cell–cell contacts prior to bundling (D). Loss of cell–cell contacts after primary bundle formation (E). Arrowheads denote the positions of cell–cell contacts. (F). Schematics of loss of cell–cell contacts after programmed disassembly of MT projections. Disassembly of MT projections involves degeneration of basal integrin-laminin complex, leading to loss of cell–cell contact. Scale bars: 5 μm (A), 10 μm (C–E).

imaging, most cells contain MT protrusions and maintain cell–cell contacts at the interepithelial space (Fig. 5A,B). During MT disassembly, MT projections start bundle formation and degeneration, thus MT distributions in the interepithelial space decrease (Fig. 5A–C). In our current protocol, quantifying the loss of cell–cell contacts is challenging. Thus, we quantified the spatial distribution of MTs during a 4-hour period. The intensities of GFP-tagged MTs localized in the interepithelial space and the peak levels of dorsal and ventral layers were measured, from which the ratio of MT distributions in the interepithelial space was calculated. This reveals that the ratio of MTs in the interepithelial space gradually decreases in a time-dependent manner (Fig. 5D).

We then counted the number of mitotic cells during a 4-hour period. Mitotic cells are detected starting around 12.5 h APF (120 min), and the number of mitotic cells reaches ~100–150 at 14.5 h APF (240 min) (Fig. 5D). As individual cells divide only once during a 4-hour window, which is consistent with previous reports (Etournay et al, 2016), ~30–35% of the observed cells undergo mitosis. Importantly, the number of mitotic cells between dorsal and ventral layers is tightly coordinated (Fig. 5D). These results suggest that MTs localized in the interepithelial space gradually decrease by release of cell–cell contacts, accordingly leading to an increasing number of mitotic cells in a coordinated manner between the two epithelia (Fig. 5C,D).

## Modulating MT stability affects coordinated mitoses

Having demonstrated that disassembly of the MT projections is tightly coupled with coordinated mitoses, we hypothesized that modulating MT stability may impact coordinated mitoses. If mitosis presupposes loss of protrusions, stabilizing or destabilizing MTs may affect mitosis. By employing conditional ectopic expression of the MT-stabilizing factor human Tau (hTau) in both dorsal and ventral pupal wing epithelia (Wang and Mandelkow, 2016), we investigated how the MT projections are regulated. Our data show that the dynamics of dorsal and ventral MT projections are less active than that of control, and the complexity of the interepithelial space perdures longer, indicating that stabilized MTs are more resistant to disassembly (Fig. 6A–A"). Importantly, the number of mitotic cells significantly decreases in both cell layers, resulting in smaller adult wings (Figs. 6A" and EV3A,C). These results suggest that the dynamic changes of MT projections during the 4-hour period play key roles in coordinated mitoses of pupal wing epithelia.

Next, we ectopically expressed hTau only in the dorsal compartment using the *apterous-Gal4 (ap-Gal4)* driver, in which Gal4 is expressed under the control of the dorsal compartment-specific *apterous* enhancer (Calleja et al, 1996). Surprisingly, we found that the number of mitotic cells was not significantly affected in either dorsal or ventral epithelia (Fig. 6B"). Correspondingly, the intensity of interepithelial MTs decreases in a time dependent manner (Fig. 6B"). Why was the hTau ectopic expression phenotype in dorsal cells not more prominent? Although hTau overexpression phenotypes persist due to slower disassembly of dorsal MT projections than ventral ones (Fig. EV3B,D,E), we found that both dorsal and ventral MT projections have similar dynamics to the control, and consistently, the complexity of the interepithelial space changes similarly (Fig. 6B–B"). These results suggest that the interepithelial space of MT projections acts as a regulator to control cell–cell contacts. When one of the cell layers is wild type, disassembly of MT projections leads to the degeneration of ECM, which seems to be sufficient to release the cell–cell contacts (Fig. EV3E').

If disassembly of the IPAN is the prerequisite for mitosis, induced disruption of MT projections may lead to increased mitoses. To understand whether this is the case, we attempted ectopic expression of MT-severing factor Katanin-60 (Kat60) in the dorsal compartment (Mao et al, 2014). This allows us to investigate the significance of MT dynamics under autonomous and non-autonomous conditions. The ectopic expression of Kat60 largely disrupted MT projections in the dorsal compartment at 10.5 h APF (Fig. 6C). In contrast, a large number of incomplete MT protrusions are observed in the ventral compartment (Fig. 6C). However, these ventral protrusions are disorganized. Therefore, the interepithelial space does not include the cell–cell contact-mediated complexed structure (Fig. 6C,C'). Our time-lapse imaging reveals that the number of mitotic cells remains very low in both the dorsal and ventral tissue (Fig. 6C"), resulting in a blistered adult wing that is smaller than in controls (Fig. EV3F). These results suggest that the formation of cell–cell contact in the interepithelial space and its subsequent loss play a central role in coordinated mitoses.

## Patronin and Short stop sustain MT-based protrusions and coordinated mitosis

As shown above, MT-based protrusions extended vertically and basally from the apical region of the cell body at 10.5 h APF at 29 °C (or 13 h APF at 25 °C, Fig. 1). Previous work has shown that vertical MT projections are nucleated by ncMTOCs in differentiated epithelial cells (Muroyama and Lechler, 2017; Röper, 2020). We thus asked whether ncMTOCs are utilized for MT nucleation in MT-based protrusions. Patronin, an ortholog of human CAMSAP2 and tubulin-binding factor at the minus end of MTs, has been characterized as one of the key components of ncMTOCs

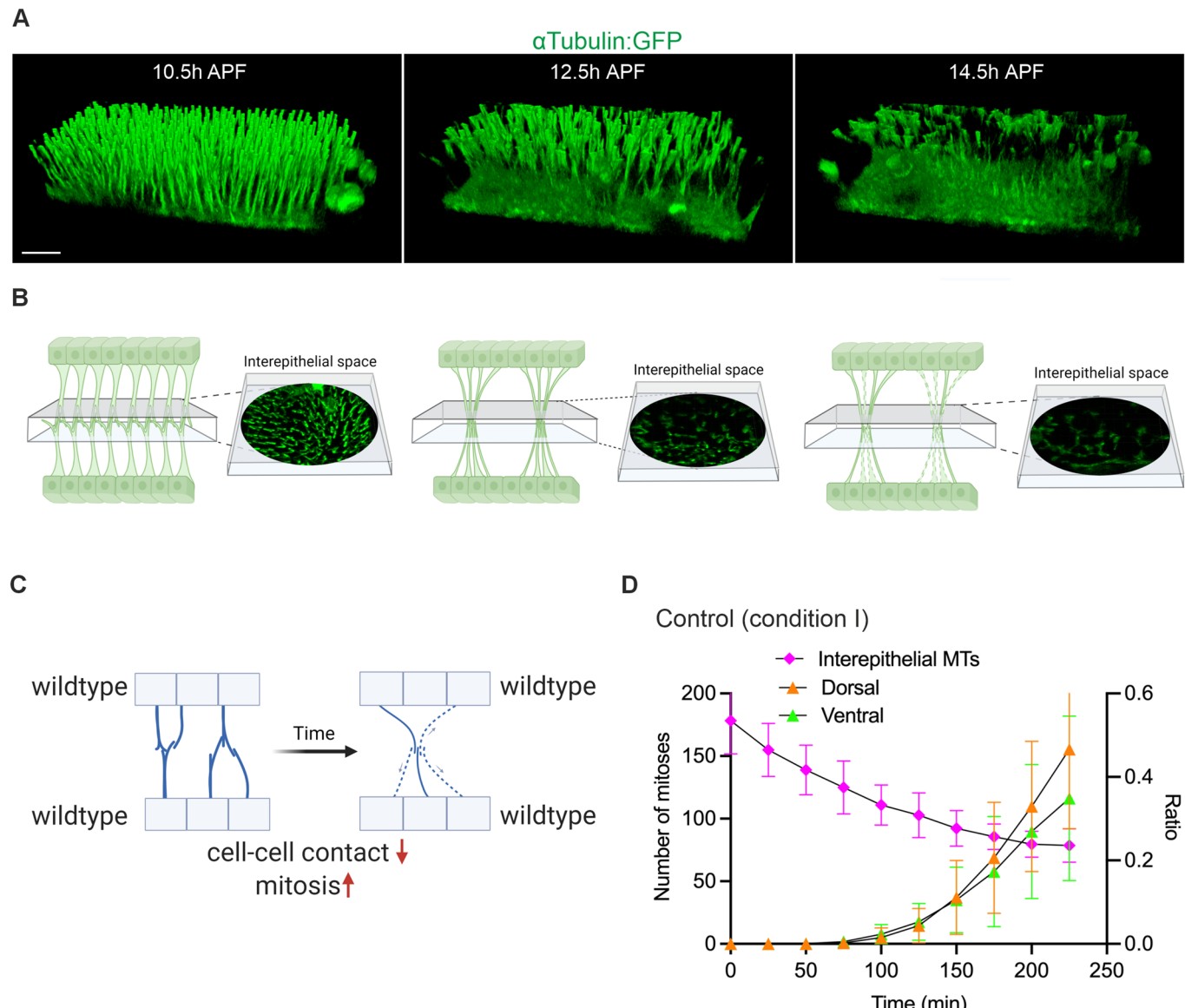

**Figure 5. Disassembly of MT protrusions and subsequent mitosis take place in a coordinated manner between dorsal and ventral epithelia.**

(A) Time lapse images of 3D view of MT protrusions visualized by αTubulin:GFP at 10.5, 12.5, and 14.5 h APF (29 °C) of the wing in control (condition I). Apical surface of the dorsal epithelium is towards the top of the view. (B) Schematics show 3D structural changes of the IPAN over time. Cuboids at left in each schematic shows approximate positions at which live images were generated. Interepithelial areas are used for quantification to measure cell–cell contacts. (C) Schematics of cell–cell contacts in both dorsal and ventral cells in control condition. (D) Number of mitotic cells (dorsal: orange triangle, ventral: green triangle) in wing epithelium and ratio of interepithelial MTs (magenta) at different time points in control pupal wings. Time 0 corresponds to 10.5 h APF. Data are from five individual replicates (N = 5). Data are means ± 95% CIs. Scale bar: 20 μm. Source data are available online for this figure.

(Akhmanova and Hoogenraad, 2015; Muroyama and Lechler, 2017; Wu and Akhmanova, 2017). When GFP-tagged Patronin was co-expressed with mCherry-labeled αTubulin, Patronin is specifically localized at the apical end of MT projections at 14 h APF (25 °C) (Fig. 7A), suggesting that MT projections at 13–14 h APF, 25 °C (or 10.5-11.5 h APF, 29 °C) are sustained by ncMTOCs.

We then studied conditional knockdown of *patronin* in both dorsal and ventral pupal wings. First, we confirmed that our knockdown protocol sufficiently reduces Patronin:GFP (Fig.

EV4A). We found that *patronin* RNAi shows significant reduction in the number of apical foci of the MT projections at 10.5 h APF, although cell–cell contact between the two epithelia is still largely maintained (Figs. 7B and EV4F). Furthermore, we observed that after *patronin* knockdown in both the dorsal and ventral layers, the number of mitotic cells is significantly reduced (Figs. 7C–C" and EV4B in control). We then asked how the dynamics of the MT projections are regulated. Our data reveal that disassembly of both dorsal and ventral projections appear to be delayed, thus the MT

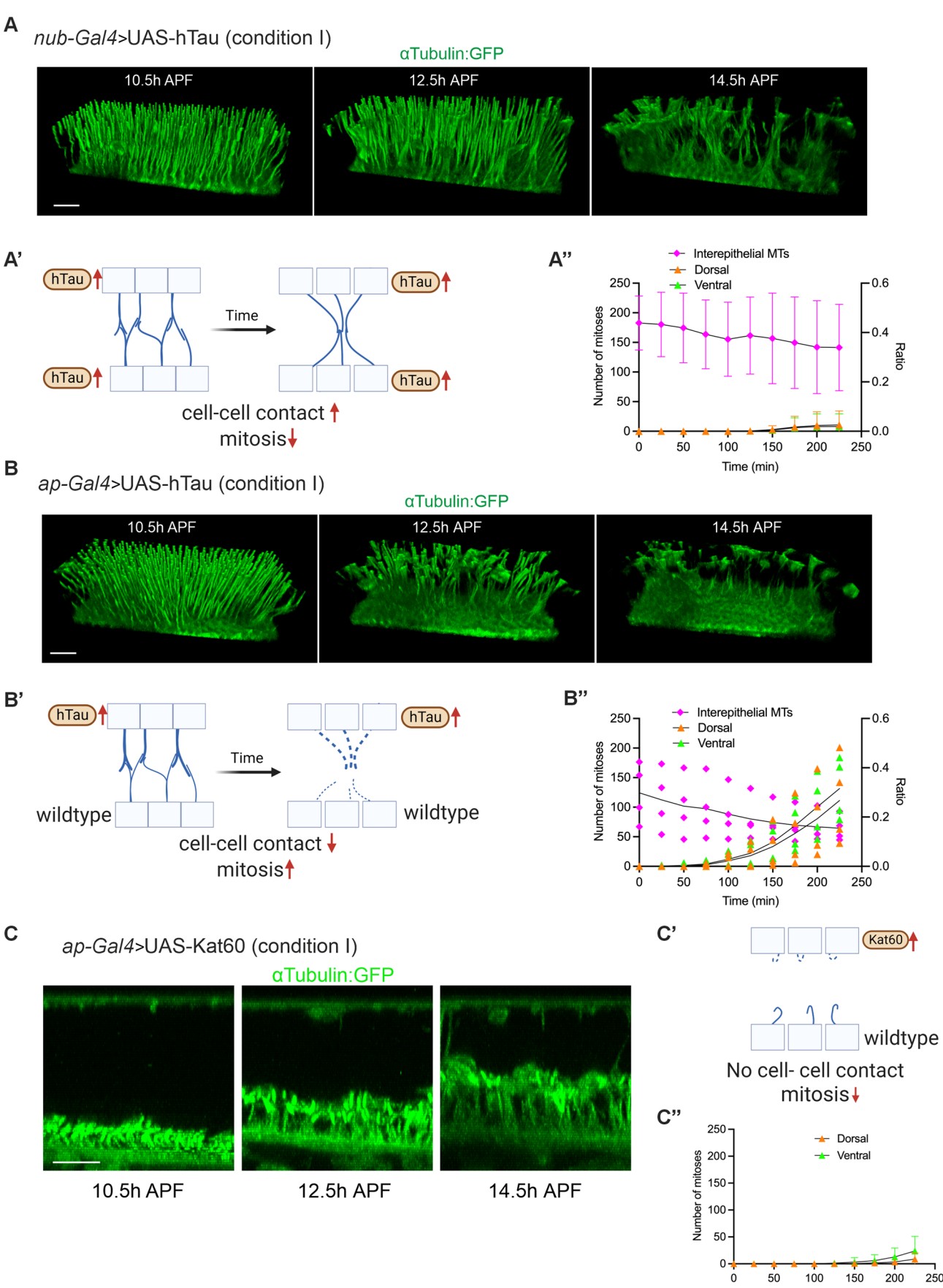

**A** *nub-Gal4*>UAS-hTau (condition I)

αTubulin:GFP

10.5h APF    12.5h APF    14.5h APF

**B** *ap-Gal4*>UAS-hTau (condition I)

αTubulin:GFP

10.5h APF    12.5h APF    14.5h APF

**C** *ap-Gal4*>UAS-Kat60 (condition I)

αTubulin:GFP

10.5h APF    12.5h APF    14.5h APF

**Figure 6. Modulating MT stability affects coordinated mitoses.**

(A) Time lapse images of 3D view of MT protrusions visualized by αTubulin:GFP at 10.5, 12.5, and 14.5 h APF (29 °C) of the wing overexpressing hTau in both dorsal and ventral cells (*nub-Gal4> hTau*). Apical surface of the dorsal epithelium is towards the top of the view. (A') Schematics of cell–cell contacts during hTau overexpression in both dorsal and ventral cells. (A") Number of mitotic cells (dorsal: orange triangle, ventral: green triangle) in wing epithelium and ratio of interepithelial MTs (magenta) at different time points during hTau overexpression (both dorsal and ventral) pupal wings (right). Time 0 corresponds to 10.5 h APF. Data are from five individual replicates ($N = 5$). Data are means ± 95% CIs. (B) Time lapse images of 3D view of MT protrusions visualized by αTubulin:GFP at 10.5, 12.5, and 14.5 h APF (29 °C) of the wing overexpressing hTau in only dorsal cells (*ap-Gal4> hTau*). Apical surface of the dorsal epithelium is towards the top of the view. (B') Schematics of cell–cell contacts of hTau overexpression only in dorsal cells. (B") Number of mitotic cells (dorsal: orange triangle, ventral: green triangle) in wing epithelium and ratio of interepithelial MTs (magenta) at different time points in hTau overexpression (dorsal) pupal wings (right). Time 0 corresponds to 10.5 h APF. $N = 4$. Data are individual replicates with means. (C) Time lapse images of 3D view of MT protrusions visualized by αTubulin:GFP at 10.5, 12.5, and 14.5 h APF (29 °C) of the wing overexpressing Kat60 in only dorsal cells (*ap-Gal4>Kat60*). Apical surface of the dorsal epithelium is towards the top of the view. (C') Schematics of cell–cell contacts of Kat60 overexpression in only dorsal cells. (C") Number of mitotic cells (dorsal: orange triangle, ventral: green triangle) in wing epithelium at different time points in Kat60 overexpression (dorsal) pupal wings (right). Time 0 corresponds to 10.5 h APF. Data are from five individual replicates ($N = 5$). Data are means ± 95% CIs. Scale bars: 20 μm (A–C). Source data are available online for this figure.

distribution in the interepithelial space persists longer than in control (Fig. 7C'). This suggests that loss of cell–cell contacts is less efficiently facilitated, thereby leading to a lower number of mitotic cells in both dorsal and ventral tissue (Fig. 7C").

We next asked how coordinated mitoses are regulated when *patronin* is knocked down only in dorsal cells using the *ap-Gal4* driver. Our data reveal that the number of mitotic cells is not significantly affected in either the dorsal or ventral epithelia even though the loss of *patronin* phenotypes is observed in dorsal MT projections (Figs. 7D–D" and EV4E–G). Our data reveal that dorsal and ventral MT projections are regulated similarly to control, and the MT distribution in the interepithelial space decreases in a time-dependent manner (Fig. 7D"). These findings suggest that Patronin facilitates the loss of cell–cell contact via uncharacterized mechanisms. Therefore, the loss of Patronin in both dorsal and ventral cells affects cell–cell contacts (Fig. 7C–C"). In contrast, since the integrin-laminin complex mediates cell–cell contacts that require a cell contact environment in both dorsal and ventral cells, one layer with a wild-type phenotype (and the other in which Patronin is reduced) follows the loss of cell–cell contacts (Fig. 7D'). Therefore, a compromised IPAN structure in one layer seems to be functionally rescued by the IPAN of the opposite layer to sustain coordinated proliferation.

Given the limited disruption of MT-based protrusions after conditional knockdown of Patronin (Fig. 7B), we wondered whether other MTOC-associated factors are also involved in IPAN-mediated interepithelial mitotic coordination. Thus, we tested loss of function of other co-factors of ncMTOCs. Short stop (Shot) is a multi-functional protein and the *Drosophila* ortholog of MACF1 (microtubule-actin-cross linking factor-1) in vertebrates (Voelzmann et al, 2017). Shot interacts with ncMTOCs by forming a complex with Patronin, and also interacts with the MF network (Nashchekin et al, 2016). Shot is localized in the apical region, when minus-ends of MTs are enriched medioapically, and in the basal domain at 13 h APF (25 °C) (Fig. EV4C; Sun et al, 2021). When Shot is conditionally knocked down only in the dorsal layer, MT projections are incompletely formed, and accordingly, the IPAN is largely abolished, leading to loss of dorsal-ventral contacts, which is consistent with a previous report (Fig. 7E,E'; Sun et al, 2021). Importantly, the number of mitoses in the ventral epithelium is drastically reduced in addition to the dorsal epithelium, even though Shot expression is not manipulated there (Fig. 7E"), leading to smaller and inflated adult wing (Fig. EV4D). These data further support the hypothesis that the formation of

cell–cell contacts and the subsequent loss of cell–cell contacts are required for coordinated mitoses between the two epithelia.

## The G2/M transition in the dorsal and ventral epithelia takes place autonomously

Previous studies indicate that a majority of wing cells in the prepupal stage are arrested in G2 (Milan et al, 1996). This leads us to hypothesize that the disassembly of MT projections is a crucial step for facilitating the G2/M transition. To gain insight into the involvement of the G2/M transition in the IPAN-mediated tissue proliferation, we conducted knockdown of String (Stg), encoding a Cdc25 phosphatase, which is a key regulator of the cell cycle by activating cyclin-dependent kinase 1 (CDK1) and is responsible for the G2/M transition (Glover, 2012; Milan et al, 1996). When conditional RNAi of *stg* was induced in the dorsal epithelium, a significant number of mitotic cells persists ventrally while the suppression of mitosis is evident dorsally (Fig. 8A–C). Notably, dorsal and ventral MT projections form bundled structures, and the MT distribution in the interepithelial space decreases, albeit slightly delayed compared to the control, resulting in a lower number of mitotic cells than the control (Figs. 8A,C and EV4B). Nonetheless, these results indicate that the G2/M transition induced by IPAN-mediated MT dynamics takes place independently within the two epithelia.

To further confirm that the G2/M transition occurs independently within the dorsal and ventral epithelia, we employed a conditional knockdown of POLO kinase, an evolutionarily conserved cell cycle regulator that controls the G2/M transition by regulating the activation of Cdk1 and by coordinating the mitotic events (e.g. spindle assembly, chromosome segregation and cytokinesis) (Glover, 2012). When Polo was conditionally knocked down in the dorsal epithelium, the dorsal layer shows loss of mitosis, but the ventral reveals mitotic activity (Fig. EV5A,B). Taken together, these results suggest that the loss of cell–cell contacts serves as proliferative signal in both dorsal and ventral epithelia, and executors of the G2/M transition thus function autonomously.

## Discussion

Here, our data show that a unique cellular mechanism mediated by the IPAN plays a key role in 3D morphogenesis during pupal wing development. Although such cellular structures have been

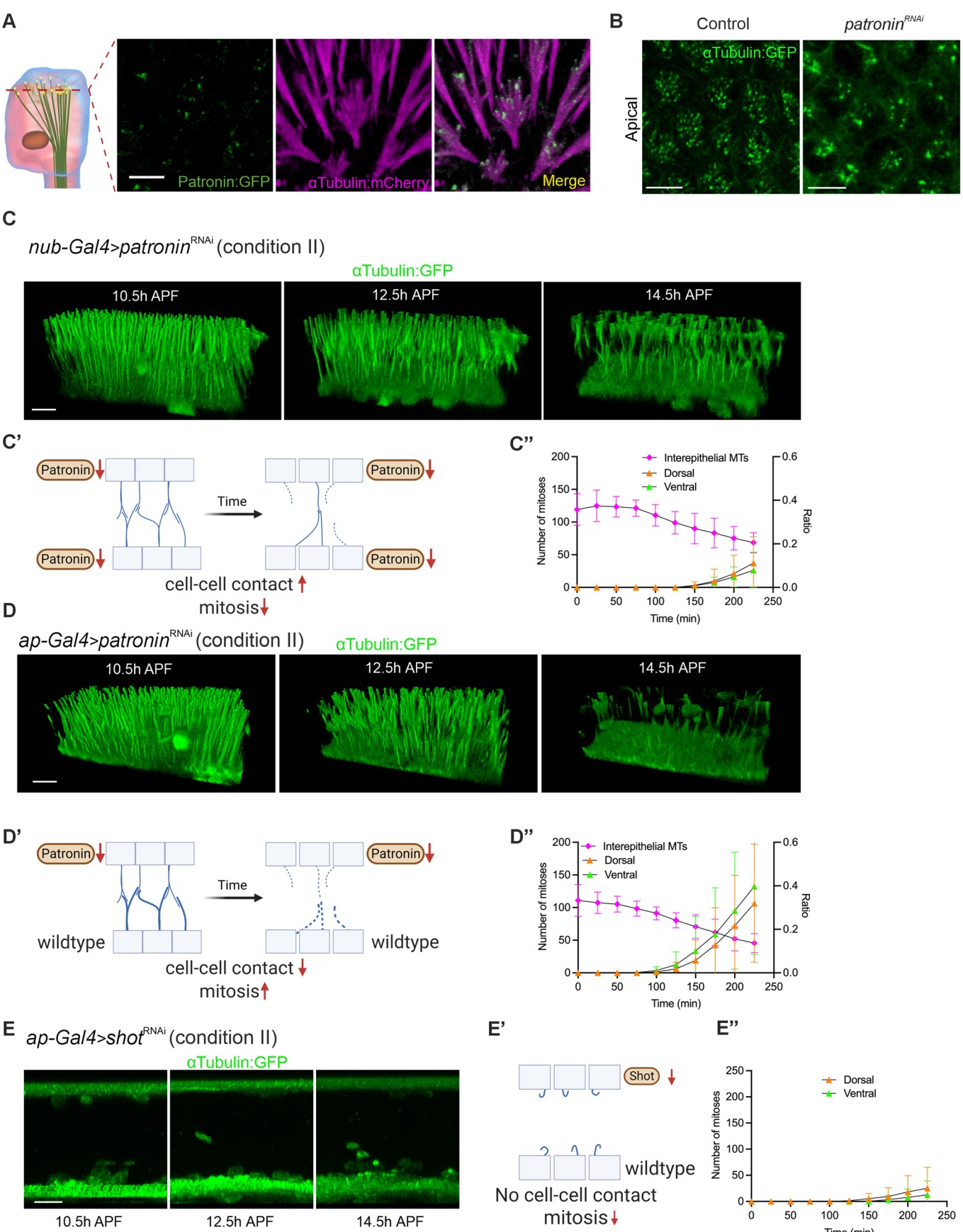

**Figure 7. MT protrusions of the IPAN are nucleated by ncMTOCs to sustain mitosis of 3D tissue.**

(A) Live-imaging of αTubulin:mCherry and Patronin:GFP at 14 h APF (25 °C). Patronin:GFP (left), αTubulin:mCherry (center), merged image (right) and schematic showing Patronin (yellow dots) and MT protrusions (green) in a single cell (far left). Patronin localizes to minus-ends of MT protrusions. (B) Apical view of αTubulin:GFP at 10.5 h APF (29 °C) in control (left) and conditional RNAi knockdown of *patronin* (right). Note that the reduced number of apically oriented MT foci upon *patronin* knockdown. (C) Time lapse images of 3D view of MT protrusions visualized by αTubulin:GFP at 10.5, 12.5, and 14.5 h APF (29 °C) of the wing in conditional *patronin* knockdown in both dorsal and ventral cells (*nub-Gal4> patronin RNAi*). Apical surface of the dorsal epithelium is towards the top of the view. (C') Schematics of cell–cell contacts during *patronin* RNAi in both dorsal and ventral cells. (C″) Number of mitotic cells (dorsal: orange triangle, ventral: green triangle) in wing epithelium and ratio of interepithelial MTs (magenta) at different time points in *patronin* RNAi (both dorsal and ventral) pupal wings (right). Time 0 corresponds to 10.5 h APF. Data are from five individual replicates (*N* = 5). Data are means ± 95% CIs. (D) Time lapse images of 3D view of MT protrusions visualized by αTubulin:GFP at 10.5 h, 12.5 h and 14.5 h APF (29 °C) of the wing during conditional knockdown of *patronin* only in dorsal cells (*ap-Gal4> patronin RNAi*). Apical surface of the dorsal epithelium is towards the top of the view. (D') Schematics of cell–cell contacts of *patronin* RNAi only in dorsal cells. (D″) Number of mitotic cells (dorsal: orange triangle, ventral: green triangle) in wing epithelium and ratio of interepithelial MTs (magenta) at different time points in *patronin* RNAi (dorsal) pupal wings (right). Time 0 corresponds to 10.5 h APF. Data are from five individual replicates (*N* = 5). Data are means ± 95% CIs. (E) Time lapse images of cross section of MT protrusions visualized by αTubulin:GFP at 10.5, 12.5, and 14.5 h APF (29 °C) of the wing during conditional knockdown of *shot* only in dorsal cells (*ap-Gal4> shot RNAi*). Apical surface of the dorsal epithelium is towards the top of the view. (E') Schematics of cell–cell contacts of *shot* RNAi only in dorsal cells. (E″) Number of mitotic cells (dorsal: orange triangle, ventral: green triangle) in wing epithelium at different time points in *shot* RNAi pupal wings (right). Time 0 corresponds to 10.5 h APF. Data are from five individual replicates (*N* = 5). Data are means ± 95% CIs. Scale bars: 5 µm (A, B), 20 µm (C, D, E). Control under condition II: Figure EV4B. Source data are available online for this figure.

described previously, their physiological significance largely remains to be addressed (Fristrom et al, 1993; Sun et al, 2021). Our in vivo live observations reveal the unique characteristics of the structure: MTs and MFs form basal vertical protrusions, and dynamics of an MF-mediated lateral filopodia-like network result in bundling of vertical protrusions. These phenomena evoke amidakuji (ghost leg), hence our term **I**nter**p**lanar **A**mida **N**etwork (IPAN) for the structure (Fig. 1F, Movie EV1). Moreover, our findings elucidate the physiological significance of the IPAN as follows. First, basal protrusions of the IPAN sustain cell–cell contact between dorsal and ventral epithelia during the early inflation stage. Second, during disassembly of the IPAN, MT projections form a bundle, leading to two outcomes: either a higher-order bundle is formed to generate a basal structure, or cell–cell contacts are lost through complete regression of MT projections. Among the cells whose MT projections regress, MTs reorganize to form mitotic spindles when cells progress to mitosis. Loss of cell–cell contact affects mitosis in both dorsal and ventral layers, supporting 3D tissue growth (Fig. 9A–D). Our data further indicate that MT projections, which are initially supported by ncMTOCs, regress, resulting in cMTOC-mediated mitotic spindle formation for mitosis. Therefore, the IPAN provides a unique framework that coordinates 3D tissue development.

## The IPAN forms an intercellular network that coordinates 3D morphogenesis

High-resolution imaging of MT protrusions shows that MTs extend from the medioapical region to the basal tip of the protrusions in the interepithelial space. We consider the structures of MT protrusions in and of themselves to be unique. MT minus-ends are located medioapically, and the number of foci is on average ~30/cell (Fig. EV1B). Moreover, MT protrusions are derived from both dorsal and ventral epithelia, and meet in the interepithelial space. To our knowledge, such bilateral MT protrusions have not been characterized in developing tissue to date. Although the detailed mechanisms of IPAN-mediated signal exchange between the two epithelia and the loss of interepithelial contact that leads to coordinated growth remain to be addressed, we hypothesized that both dorsal and ventral protrusions play instructive and permissive roles (Figs. 6, 7, and 9). The following are the conditions under which IPAN-mediated coordinated

mitoses occur: First, the timing of IPAN disassembly is important, as delaying it significantly affects tissue growth (Figs. 5 and 6). Second, cell–cell contact before disassembly is crucial: Loss of *shot* or ectopic expression of Kat60 in dorsal tissue induces disruption of cell–cell contact, which sufficiently affects growth of both dorsal and ventral tissues, even when ventral cells are wild type (Figs. 6 and 7). Third, MT protrusions are fully functional for coordinated mitoses only when they are nucleated by Patronin-mediated ncMTOCs, however, these processes can be rescued by the other cell layer, if cell–cell contact is maintained (Fig. 7). Therefore, the IPAN does not simply antagonize mitosis, but rather serves as an instructive mitotic coordinator during MT projection disassembly.

Coupling between loss of MT protrusions and mitosis has been described in primary cilia in vertebrate cells (Plotnikova et al, 2009; Pugacheva et al, 2007). We consider the IPAN-mediated mitosis to be distinct from cilia-mediated mitosis: primary cilia are linked to centrosomes and are nucleated by cMTOCs, thus suppressing mitotic spindle formation (Plotnikova et al, 2009).

One of the distinctive features of the IPAN is the presence of lateral filopodia-like structures between vertical protrusions (Figs. 1E,F and EV1C). Intercellular networks mediated by filopodia-like structures in developing tissue have been previously described that determine cell fate in the *Drosophila* embryo and direct morphogenesis in the quail embryo (Sato et al, 2017; Zhang et al, 2018). Our time-lapse imaging focused on the lateral network reveals that dynamics of filopodia-like structures facilitate bundle formation from vertical protrusions (Fig. 2B,C, Movie EV3). Similar structures appear to be commonly used for intercellular networks in developing tissues. Further studies are needed to address the physiological significance of the filopodia-like structures of the IPAN.

## MTs shuttle between ncMTOCs and cMTOCs

One of the key findings in this work is that MTs are restructured from ncMTOCs to cMTOCs during the inflation stage of pupal wing development. During the early inflation stage, e.g. 13 h APF (25 °C), the majority of MTs are utilized for vertical protrusions and are nucleated by ncMTOCs at minus-ends of MTs in the medioapical region (Fig. 1D). Previous studies indicated that ncMTOC-regulated MTs are used in differentiated cells, e.g.

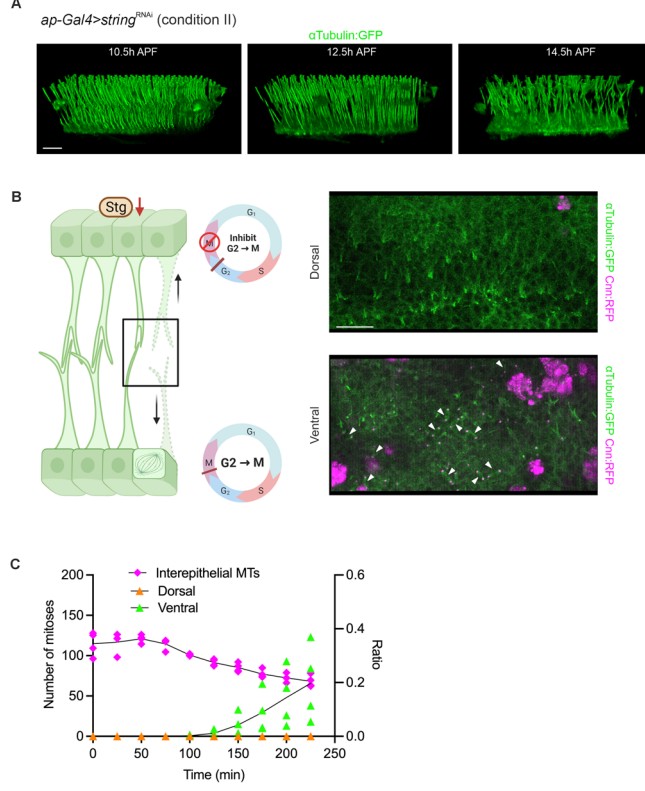

**Figure 8.  G2/M executor String functions autonomously.**

(A) Time lapse images of 3D view of MT protrusions visualized by αTubulin:GFP at 10.5, 12.5, and 14.5 h APF (29 °C) of the wing during conditional knockdown of *string* only in dorsal cells (*ap-Gal4>string RNAi*). Apical surface of the dorsal epithelium is towards the top of the view. (B) Left: Schematic of cell–cell contacts and mitosis of *string* RNAi only in dorsal cells. After the loss of cell–cell contacts (black rectangle), G2/M execution signals transfer to both dorsal and ventral cells (arrows), but only ventral cells undergo mitosis because of the loss of *stg* in dorsal cells. Right: Dorsal (top) and ventral (bottom) epithelial cells visualized by αTubulin:GFP (green) and Cnn:RFP (magenta) at 14.5 h APF. Note that mitotic cells are only observed in ventral epithelium (arrowheads). (C) Number of mitotic cells (dorsal: orange triangle, ventral: green triangle) in wing epithelium and ratio of interepithelial MT (magenta) at different time points in *string* RNAi pupal wings. Time 0 corresponds to 10.5 h APF. N = 4. Data are individual replicates with means. Scale bars: 20 μm (A, B). Source data are available online for this figure.

neurons and epithelial organs, after termination of growth phase (Booth et al, 2014; Gillard et al, 2021; Muroyama and Lechler, 2017; Toya et al, 2016; Wu and Akhmanova, 2017). In contrast, this study suggests that the IPAN, a transient cellular structure sustained by ncMTOC-nucleated MTs, provides a pool of tubulin subunits upon its disassembly that is utilized for mitotic spindle MT formation nucleated by cMTOCs (Fig. 4, Movie EV6). Therefore, organization of MTs can shuttle between ncMTOCs and cMTOCs in a context-dependent manner. Although, to our knowledge, restructuring MTs from ncMTOCs to cMTOCs has not been characterized in epithelial tissue, dynamic MT restructuring occurs in some physiological and pathological conditions, e.g. epithelial–mesenchymal transition (EMT) and neoplasia of epithelial tissue (Burute et al, 2017; Schnerch and Nigg, 2016).

## Significance of membrane protrusions is a frontier field in tissue development

Although membrane protrusions themselves have been occasionally described in developing tissues (Korenkova et al, 2020), genetic manipulation focused on protrusions is often challenging. Therefore, current knowledge of the physiological significance of such structures is limited.

Our proposed model reveals that the loss of cell–cell contacts functions as a key regulator of coordinated mitoses (Fig. 9D). Visualizing tools to monitor cell–cell contact have been developed (Cabantous et al, 2005; Hu and Kerppola, 2003), but technologies to quantify the loss of cell–cell contacts have yet to be created. Such an in vivo developmental system will allow us to investigate coordinated mitoses through the loss of cell–cell contacts with more precision.

Nevertheless, this study of the IPAN provides several useful insights in the field. First, our in vivo live imaging protocol is relatively simple and can be carried out using confocal microscopy. Additionally, various cell shapes in epithelia, including those observed in membrane protrusion- positive or -negative cells and in cells undergoing mitosis, as well as MT dynamics such as ncMTOC-nucleated vertical protrusions, can be observed in a relatively short time frame. Moreover, powerful *Drosophila* genetics tools, e.g. the Gal4/Gal80 system of spatiotemporally controlled gene expression, are straightforward to use in combination with multi-colored fluorescent in vivo live-imaging. Thus, the convenient and comprehensive system of the IPAN in the *Drosophila* pupal wing shows promise in addressing many questions about cell shape changes impacting tissue morphogenesis that were previously difficult to answer.

## Future perspectives

Although this study shows an unconventional cellular mechanism involving loss of cell–cell contacts leading to coordinated tissue growth, several key questions regarding the IPAN remain to be addressed.

First, our data reveal that disassembly of the IPAN is key for coordinated growth. However, such molecular mechanisms remain to be addressed. Although the majority of future intervein cells contain membrane protrusions around 13 h APF at 25 °C, how the loss of protrusions is spatiotemporally regulated remains unknown.

Second, one of the key findings in this study is that regulation of MT nucleation shuttles between different types of MTOCs during tissue development. The molecular mechanisms that involve such changes in MT nucleation thus remain to be addressed. This is crucial not only for understanding tissue morphogenesis, but also for investigating pathological conditions, such as neoplasia arising in epithelia.

Third, time lapse imaging of membrane-bound forms of fluorescent proteins shows that vesicle trafficking appears to take place within the IPAN structure (Movie EV3). How vesicle trafficking is regulated, what components are transported (e.g. organelles such as mitochondria or endosomes), whether transported components play a significant role in tissue development, and whether trafficking goes through or beyond cell boundaries are all questions that arise by observing in vivo live images during tissue development. Tissue culture experiments in which tunneling

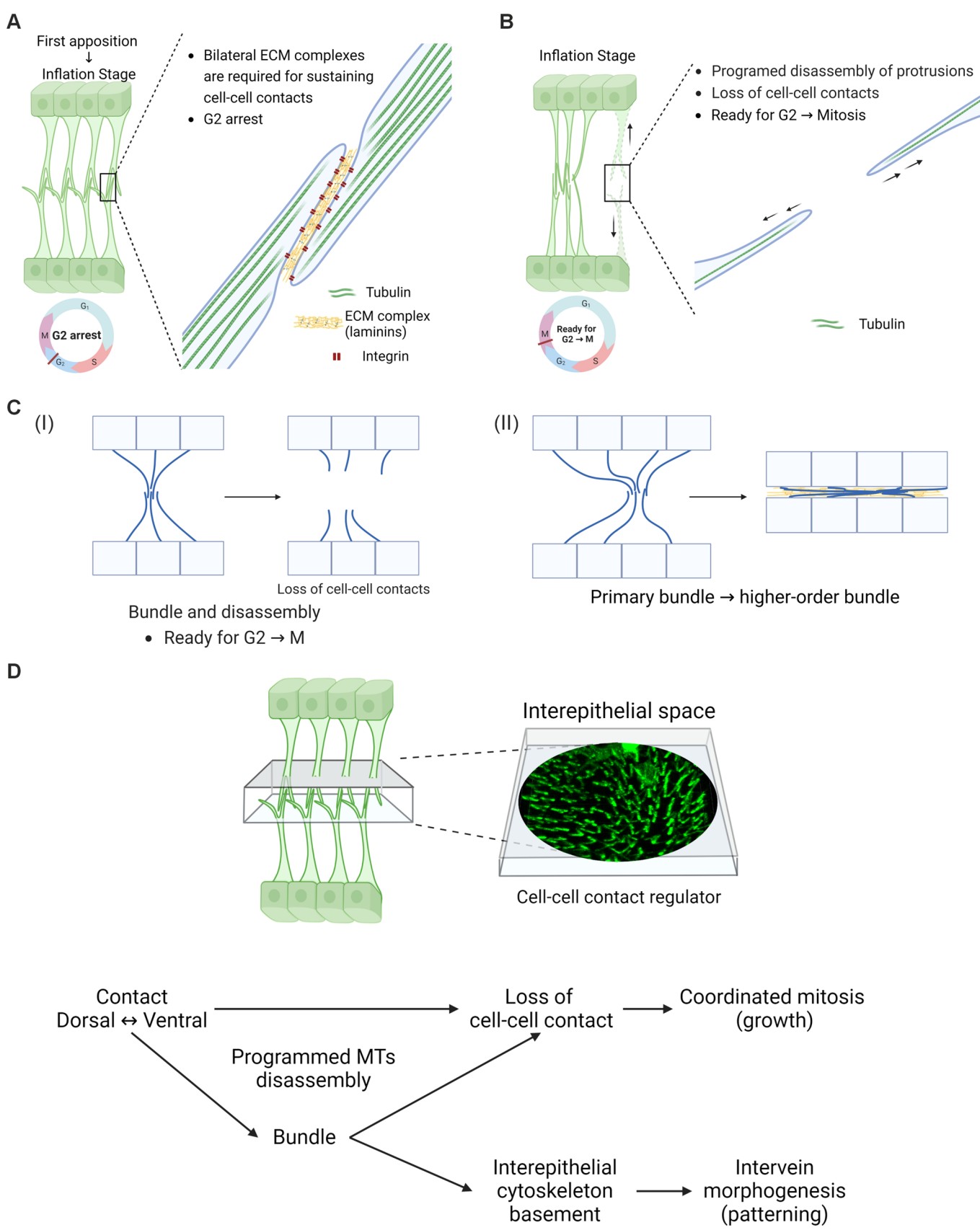

**Figure 9. Schematics of IPAN-mediated 3D morphogenesis during pupal wing development.**

(A) During early inflation stage, basally branched MT protrusions sustain cell–cell contacts between the two epithelia through interactions of basal integrin-laminin complex. Wing epithelial cells largely remain in G2 arrest. (B) After head eversion around ~13-14 h APF (25 °C), disassembly of MT projections involves degeneration of ECM, leading to loss of cell–cell contacts, which are required for releasing cells from G2 arrest. (C) During MT disassembly, the majority of basal protrusions starts bundling with neighboring protrusions, forming primary bundles. Protrusions within some primary bundles undergo disassembly, resulting in loss of cell–cell contacts that facilitate coordinated mitoses (i). Some of the primary bundles further aggregate into higher-order bundles that form an interepithelial cytoskeletal basement prior to the second apposition (ii). (D) Summary of the IPAN dynamics. Interepithelial space mediating cell–cell contact serves as a regulator of coordinated mitoses between the two epithelia. Two fates after a primary bundle formation, either disassembly of protrusions or higher-order bundling, lead to coordinated mitoses for tissue growth, or interepithelial cytoskeletal basement formation for intervein morphogenesis, respectively.

nanotubes were first characterized have shed light on the significance of such cell–cell contact-mediated trafficking (Rustom et al, 2004). Interestingly, tunneling nanotubes serve as a platform of intercellular trafficking pathogens as well (Kumar et al, 2017; Scheiblich et al, 2021; Zhang et al, 2021). Hence, future study of the IPAN may contribute to understanding how trafficking through it is utilized in physiological conditions during tissue development, as well as under pathological conditions.

In summary, our data reveal a novel cellular mechanism that coordinates 3D morphogenesis. We postulate that such a mechanism may not only be in play in *Drosophila* wing development, but may also serve as a system for 3D morphogenesis in development throughout the animal kingdom.

# Methods

## Fly genetics

*w; ap-Gal4* (#3041), *w; nub-Gal4* (#25754), *w; ap-lexA* (#54268), *w; ubi>patronin:GFP/CyO* (#55128), *UAS-Kat60* (#64117), *UAS-shot* RNAi (#64041), *UAS-stg* RNAi (#34831), *UAS-polo* RNAi (#33042), *UAS-Shot:GFP* (#29042), *w;; tub-Gal80^ts^* (#7017), *UAS- αTubulin:mCherry* (#25744), *UAS-hTau* (#64389), *UAS-LifeAct:Ruby* (#35545), *UAS-CAAX:mCherry* (#59021) were obtained from the Bloomington *Drosophila* Stock Center (BDSC). *UAS-Patronin* RNAi (#27654) was obtained from Vienna *Drosophila* Resource Center. *actp-S/G2/M-Green* (#109678) was obtained from Kyoto Drosophila Stock Center. *w, ubi>αTubulin:GFP* and and *ubi>RFP: αTubulin* was from C. Gonzalez (Rebollo et al, 2004), *w; ubi> cnn:RFP/SM5, CyO* from J.W. Raff (Basto et al, 2008) and *act>y>LHV2* from K. Basler (Yagi et al, 2010).

All stocks were maintained either at room temperature (21–22 °C) or in a 25 °C incubator. To generate pupal wings for imaging, crosses were set up and maintained at room temperature to silence RNAi knockdown/ectopic transgene expression phenotypes. For crosses with flies in which transgenes were ectopically expressed, white pupae were collected at room temperature, then shifted to 29 °C until pupae were aged to 10.5 h APF (equivalent to 13.5 h APF at 25 °C), ensuring pupal specific gene expression (condition I). Pupae from crosses with RNAi transgene-bearing flies were shifted to 29 °C 16 h prior to collecting white pupae, which in turn were kept at 29 °C until their imaging commenced at 10.5 h APF, ensuring pupal specific RNAi phenotypes (condition II). Head eversion was used as a developmental marker for assessing suitability for live imaging. Male pupae were selected for imaging due to slightly smaller size (and thus reduced distance

between dorsal and ventral wing epithelia), and to maximize αTubulin:GFP expression due to dosage compensation (the *ubi>αTubulin:GFP* transgene is an X-chromosome insertion).

## Full genotypes

Figure 1C,D: *w, ubi>αTubulin:GFP (X)*

Figure 1E: *w, ubi>αTubulin:GFP; nub-Gal4, UAS-LifeAct:Ruby*

Figure 1G: *w; nub-Gal4, UAS-LifeAct:GFP/UAS-CAAX:mCherry; tub>Gal80^ts^/+*

Figure 2A: *w; nub-Gal4, UAS-CAAX:mCherry*

Figure 2B,C: *w, ubi>αTubulin:GFP; nub-Gal4, CAAX:mCherry*

Figure 2D: *w, ubi>αTubulin:GFP (X)*

Figure 3A–D: *w; ap-lexA, lexAop-rCD2.RFP, lexAop-GFPi/nub-Gal4, UAS-mCD8:GFP; lexAop-FLP, act>y>LHV2/TM6B/+*

Figure 4A,C–E: *w, ubi>αTubulin:GFP; ubi>cnn:RFP/CyO; tub>Gal80^ts^/+*

Figure 5A: *w, ubi>αTubulin:GFP; ubi>cnn:RFP/CyO; tub>Gal80^ts^/+*

Figure 6A: *w, ubi>αTubulin:GFP; nub-Gal4, ubi>cnn:RFP/UAS-hTau; tub>Gal80^ts^/+*

Figure 6B: *w, ubi>αTubulin:GFP; ap-Gal4, ubi>cnn:RFP/UAS-hTau; tub>Gal80^ts^/+*

Figure 6C: *w, ubi>αTubulin:GFP; ap-Gal4, ubi>cnn:RFP/UAS-Kat60; tub>Gal80^ts^/+*

Figure 7A: *w; nub-Gal4, UAS-αTubulin:mCherry/+; ubi>Patronin:GFP/+*

Figure 7B:

control: *w, ubi>αTubulin:GFP; ubi>cnn:RFP/CyO; tub>Gal80^ts^/+*

*patronin^RNAi^*: *w, ubi>αTubulin:GFP; nub-Gal4, ubi>cnn:RFP/UAS-patronin RNAi; tub>Gal80^ts^/+*

Figure 7C: *w, ubi>αTubulin:GFP; nub-Gal4, ubi>cnn:RFP/UAS-patronin RNAi; tub>Gal80^ts^/+*

Figure 7D: *w, ubi>αTubulin:GFP; ap-Gal4, ubi>cnn:RFP/UAS-patronin RNAi; tub>Gal80^ts^/+*

Figure 7E: *w, ubi>αTubulin:GFP; ap-Gal4, ubi>cnn:RFP/UAS-shot RNAi; tub>Gal80^ts^/+*

Figure 8A,B: *w, ubi>αTubulin:GFP; ap-Gal4, ubi>cnn:RFP/UAS-string RNAi; tub>Gal80^ts^/+*

Figure EV1A: *w, ubi>αTubulin:GFP (X)*

Figure EV1C: *w, ubi>αTubulin:GFP; nub-Gal4, UAS-CAAX:mCherry*

Figure EV1D: *w; ap-Gal4; ubi > RFP:αTubulin, tub>Gal80^ts^/UAS-GFP.nls*

Figure EV2A: *w; actp-S/G2/M-Green; ubi > RFP:αTubulin*

Figure EV2B: *w, ubi>αTubulin:GFP; nub-Gal4, ubi>cnn:RFP/CyO; tub>Gal80^ts^/+*

Figure EV3A,B: *w, ubi>αTubulin:GFP; ubi>cnn:RFP/CyO; tub>Gal80^{ts}/+*

Figure EV3C,D: *w, ubi>αTubulin:GFP; nub-Gal4, ubi>cnn:RFP/ UAS-hTau; tub>Gal80^{ts}/+*

Figure EV3E: *w, ubi>αTubulin:GFP; ap-Gal4, ubi>cnn:RFP/ UAS-hTau; tub>Gal80^{ts}/+*

Figure EV3F: *w, ubi>αTubulin:GFP; ap-Gal4, ubi>cnn:RFP/ UAS-Kat60; tub>Gal80^{ts}/+*

Figure EV4A: *w; nub-Gal4, UAS-αTubulin:mCherry/UAS-patronin RNAi; ubi>Patronin:GFP/+*

Figure EV4C: *w; nub-Gal4 > UAS-αTubulin:mCherry/UAS-Shot:GFP*

Figure EV4D: *w, ubi>αTubulin:GFP; ap-Gal4, ubi>cnn:RFP/ UAS-shot RNAi; tub>Gal80^{ts}/+*

Figure EV4E: *w, ubi>αTubulin:GFP; ubi>cnn:RFP/CyO; tub>Gal80^{ts}/+*

Figure EV4F: *w, ubi>αTubulin:GFP; nub-Gal4, ubi>cnn:RFP/ UAS-patronin RNAi; tub>Gal80^{ts}/+*

Figure EV4G: *w, ubi>αTubulin:GFP; ap-Gal4, ubi>cnn:RFP/ UAS-patronin RNAi; tub>Gal80^{ts}/+*

Figure EV5A,B: *w, ubi>αTubulin:GFP; ap-Gal4, ubi>cnn:RFP/ UAS-polo RNAi; tub>Gal80^{ts}/+*

## Dorsoventral bicolor stocks

Crossing the following two stocks results in pupae in which mCD8:GFP is expressed in dorsal and ventral epithelia of the pupal wing, while however mCD8:GFP expression is knocked down to nearly undetectable levels in the dorsal layer.

*w; ap-lexA, lexAop-rCD2:RFP, lexAop-GFP-RNAi /CyO; lexAop-FLP, act>y>LHV2/TM6B*

*w; nub-Gal4, UAS-mCD8:GFP*

Thus, pupal wings are obtained with rCD2:RFP expressed (and mCD8:GFP expression repressed) dorsally, and mCD8:GFP expressed ventrally, resulting in a pupal wing in which dorsal and ventral epithelia can be visualized in two different colors. *ap-lexA* is only transcribed in the larval stage but not the pupal stage; the FLP-out system was thus utilized to maintain *lexA* expression in the dorsal tissue of the pupal wing (Yagi et al, 2010).

## Imaging and image analysis

### Preparing pupal wings for live imaging

Male pupae were collected as white pupae for live imaging and aged at 29 °C as described. At 10.5 h APF, pupae were carefully rinsed in a droplet of deionized water with a paintbrush on a Sylgard dissection plate, avoiding damage to the pupal case while removing autofluorescent debris from its surface. The pupae were then carefully placed on a Kimwipe to dry for three minutes. After drying, the pupae were placed on a piece of double-sided tape in a manner that oriented the hinge region of the pupa's right wing upward. A microscalpel (Fine Science Tools, cat# 10316-14) was used to make an incision in the pupal case along the wing margin, taking care to avoid any damage to underlying cuticle and tissue. A window was generated over the wing hinge region by carefully lifting the cut edge of the incision away from the underlying tissue with the microscalpel (the pupal case cracks naturally along the pupal dorsoventral axis, facilitating the generation of the window). A final cut with the scalpel parallel to the first incision completed

the window. Using a disposable 10 µl pipet tip, a tiny droplet of halocarbon oil (SigmaAldrich, cat# H8898) was dabbed onto the surface cuticle of the exposed hinge region to prevent dehydration during subsequent imaging. A strip of the double-sided tape to which the pupa is fastened during window cutting was removed to lift the pupa, with its thin strip of tape, off the Sylgard plate, and the pupa was then taped window-side down onto a 24 × 50 mm coverslip (thickness #1).

### Live imaging of pupal wings

Live imaging was carried out using a Leica TCS SP8 STED 3X CW 3D enclosed in a temperature-controlled chamber set to 29 °C. Channels were optimized for simultaneous acquisition of EGFP and mCherry/RFP fluorescence. The reduction in fluorescent signal as a function of imaging depth was compensated for by incremental increases of laser power from one optical confocal section to the next using a *Z*-compensation function. For each pupal wing, a 2048 × 2048 pixel × 60–75 µm volume cuboid in the vicinity of the anterior cross vein (i.e. in a region destined to become wing hinge) was imaged once every 5 min for 4 h, resulting in 48 3D time-lapse "frames". Overlapping optical sections ~1 µm-thick were taken.

### Counting mitoses in D/V epithelia

Raw confocal data (saved as .lif files) were converted to .ims files in Imaris File Converter (Oxford Instruments). All subsequent steps were carried out in Imaris (Oxford Instruments).

### Mitoses

Within the 2048 × 2048 pixel × 60–75 µm cuboid, a subregion of 510 × 1025 pixels × 60–75 µm in the immediate vicinity of the anterior crossvein, in which MT protrusions were abundant at the onset of live imaging, was chosen for tracking dorsal and ventral layer mitoses and concomitant disappearance of MT protrusions. Both αTubulin:GFP and Cnn:RFP were used as mitotic markers: cortical αTubulin:GFP highlighted boundaries of cells that shifted from polygonal to round at the onset of mitosis, and also formed spindles and midbodies as a function of mitotic stages. The presence of two Cnn:RFP foci further confirmed cells as mitotic. In every 3D frame, newly arising mitoses were marked as surfaces, and each surface was copied into every subsequent frame. Thus, the sum total of surfaces in the final frame represented the total number of mitoses that had occurred in the 510 × 1025 pixel area over the four-hour imaging period. Both dorsal and ventral layer mitoses were counted. Generally, no or few mitoses were observed in either epithelial layer until 12–12.5 h APF, and the number of mitoses accelerated towards the end of the four-hour imaging period. Dorsally, mitoses were observed in a ~5 µm-thick optical slice parallel to the coverslip. The ventral epithelium is slanted, with proximal regions being closer to the dorsal epithelium than distal regions, and thus ventral mitoses were observed over a 10–12 µm-thick optical slice. As the ventral wing epithelium abuts the developing midleg, in which cells are generally larger than in the wing, wing and leg mitoses were distinguishable from each other, and leg mitoses were thus excluded from the mitotic counts.

### Quantifying microtubule distribution

For the analysis of spacial distribution of GFP-tagged MTs, we first extracted the *z*-profile of intensity for each time point by averaging each *z*-slice, and identified the *z*-positions of dorsal and ventral

layers, $z_d$ and $z_v$, and their intensity values, $I_d$ and $I_v$, respectively. We then computed the average intensity values, $I_{int}$, of the interepithelial space between $(z_d + 10)$ and $(z_v - 10)$ in the unit of $1.0409333\,\mu m$. Finally, we computed the ratio of the interepithelial value to the average of the dorsal and ventral values for each time point, as the relative expression level of MT in the interepithelial space: $2I_{int}/ (I_d + I_v)$.

For the extraction of the $z$-profiles, we note that we first performed downsampling of the 4D data in the $x$ and $y$ directions from $1025 \times 510$ pixels to $10 \times 5$ pixels, keeping the average intensity of each $z$-layer, as the size of the 4D data is too large to handle at once. The $z$-positions of two layers can be relatively easily found as the two intensity peaks separated by about $50\,\mu m$. Below is an example of the python script to process ics-ids data file and extract the ratios.

```python
# a python script: z-axis_profile.py
# note: for importing ics-ids files. pyimagej 1.0.0 and related packages are required.
# USAGE: python z-axis_profile.py base_name_of_the_ics_file
import numpy as np
import imagej
import sys, os
import matplotlib.pyplot as plt
# constant variables
Fiji_dir = '/usr/Fiji.app' #Fiji directory path
path = os.getcwd()+'/'
datatype = 'uint8' # default data type is set 'unsigned 8-bit int'
dallow = 10 # half z-height of dorsal layer
h_expected = 25 # half of the approximate gap between dorsal and ventral layers from raw data
### Read Image data of Ch0
# Set file names
name_base = path + 'base_name_of_ics_file_without_file_extension'
name_base = path + sys.argv[1] if (len(sys.argv) > 1) else name_base
infile = name_base + '.ics'
file_pic = name_base + '-tprofile.png'
file_t = name_base + '-tprofile.txt'
file_intensity_check = path + '/intensity-zprofile.txt'
# Launch ImageJ and open the file
ij = imagej.init(Fiji_dir, mode=imagej.Mode.INTERACTIVE)
#headless=False is deprecated in a new ver.
print("#Reading a file: ",infile)
ij_img = ij.io().open(infile)
# Send it to numpy
np_img = ij.py.from_java(ij_img).astype(datatype)
print(np.shape(np_img)) #(frame, z, y, x, ch)
# Convert 3 or 5 dim data to 4 dim data of (t,z,y,x) to fit to imagej
if len(np_img.shape) == 3:
np_img = np_img.reshape((1,np_img.shape[0],np_img.shape[1],np_img.shape[2]))
if len(np_img.shape) == 5:
np_img = np_img[:,:,:,:,0]
# find the z-positions of dorsal/ventral layers
# read/set fundamental parameters and variables of the image
ymax = np_img.shape[2]-1
xmax = np_img.shape[3]-1
n_z = np_img.shape[1] # no. of z-slices
zave = np.average(np_img, axis = (2,3)) # z-profile of intensity averaged
tave = np.sum(zave,axis=1) #t-profile of the total intensity
dmid = np.argmax(zave[:,:3*dallow],axis=1) # average z position of dorsal layer as the brightest z
vmid = np.zeros(dmid.shape)
vmid = vmid.astype('int64')
for i in range(dmid.shape[0]):
vmid[i] = np.argmax(zave[i,dmid[i]+h_expected:])
vmid += dmid + h_expected
print("\nAverage positions of dorsal/ventral layers are:", dmid, vmid)
print("xmax,ymax = ",xmax,ymax)
#calculate average intensity of the bulk between dorsal and ventral layers
x = list(range(dmid.shape[0]))
y1=np.zeros(dmid.shape[0])
y2=np.zeros(dmid.shape[0])
y3=np.zeros(dmid.shape[0])
result=np.zeros(dmid.shape[0])
for i in range(dmid.shape[0]):
y1[i] = zave[i,dmid[i]]
y2[i] = zave[i,vmid[i]]
y3[i] = np.average(zave[i,(dmid[i]+10):(vmid[i]+1-10)])
result[i] = 2*y3[i]/(y1[i]+y2[i])
#output: processed data and plot
plt.xlabel("Time")
plt.ylabel("Intensity Ratio")
plt.plot(x,tave/tave[0], label = "total")
plt.plot(x,y1/y1[0], label = "dorsal")
plt.plot(x,y2/y2[0], label = "ventral")
plt.plot(x,y3/y3[0], label = "middle")
plt.plot(x,result, label = "ratio")
plt.title(infile)
plt.legend()
plt.savefig(file_pic)
np.savetxt(file_intensity_check, zave.T)
np.savetxt(file_t, np.vstack((x,result,tave,y1,y2,y3)).T, delimiter = '\t', header = '#frame no, ratio of middle value, total intensity, dorsal peak val, ventral peak val, middle average value', footer = '', comments = '')
sys.exit(0)
```

## Statistics

Statistical analyses were performed using GraphPad Prism software (v.9.0.2, GraphPad). The number for all quantified data is included in the figure legends. Data are mean ± 95% confidence intervals (CIs) ($n \geq 5$) or individual replicates with mean ($n < 5$).

## Peer review information

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

## Acknowledgements

We thank Maria Vartiainen for thoughtful comments on the manuscript; C. Gonzalez, J.W. Raff, K. Basler, the Vienna *Drosophila* Resource Center (VDRC) and the Bloomington *Drosophila* Stock Center (BDSC) for fly stocks; and the Light Microscopy Unit of the Institute of Biotechnology of the University of Helsinki for their support. *Drosophila* work was supported by the Hi-Fly core facility, funded by Helsinki Institute of Life Science and Biocenter Finland. This work was supported by the Center of Excellence in Experimental and Computational Developmental Biology (grant 272280), grant 347569 from the Academy of Finland, the Sigrid Jusélius Foundation, MOBERC33 from the Estonian Research Council to O.S. Open access funded by Helsinki University Library.

## Author contributions

**Ngan Vi Tran**: Conceptualization; Investigation; Visualization; Methodology; Writing—original draft; Writing—review and editing. **Martti P Montanari**: Conceptualization; Investigation; Methodology; Writing—original draft; Writing—review and editing. **Jinghua Gui**: Conceptualization; Investigation. **Dmitri Lubenets**: Visualization; Methodology. **Léa Louise Fischbach**: Investigation. **Hanna Antson**: Investigation. **Yunxian Huang**: Investigation. **Erich Brutus**: Visualization. **Yasushi Okada**: Methodology. **Yukitaka Ishimoto**: Software; Methodology. **Tambet Tõnissoo**: Methodology. **Osamu Shimmi**: Conceptualization; Supervision; Funding acquisition; Investigation; Visualization; Methodology; Writing—original draft; Writing—review and editing.

## Disclosure and competing interests statement

The authors declare no competing interests.

# Expanded View Figures

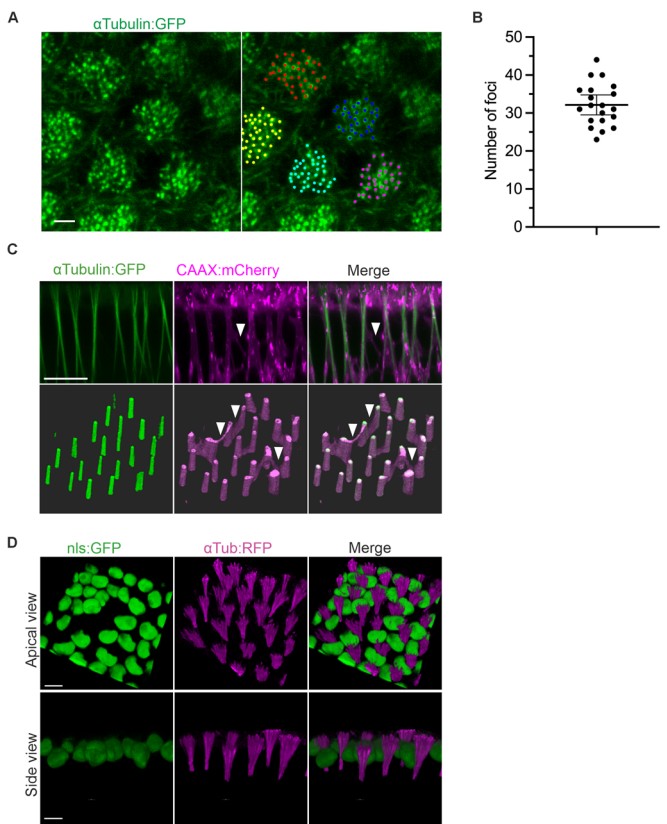

**Figure EV1. Detailed structure of the IPAN.**

(A) Apical view of MTs (left; green, αTubulin:GFP). Numbers of MT foci are counted (right; dots in various colors). (B) Number of foci of MTs in the apical compartment of each cell in (A). $n = 20$. Data are means ± 95% confidence intervals (CIs). (C) Both the vertical MT protrusions (green, αTubulin:GFP) and the horizontal MFs that connect them are enveloped in cell membrane (magenta, CAAX:mCherry). Upper panels provide an optical cross-sectional, and lower, an oblique view (apical towards the top of the page). White arrowheads point at lateral filopodia-like structures. (D) Cell nuclei are interspersed among the MT protrusions. Scale bars: 1 µm (A), 10 µm (upper panel in C), 5 µm (D).

**A**

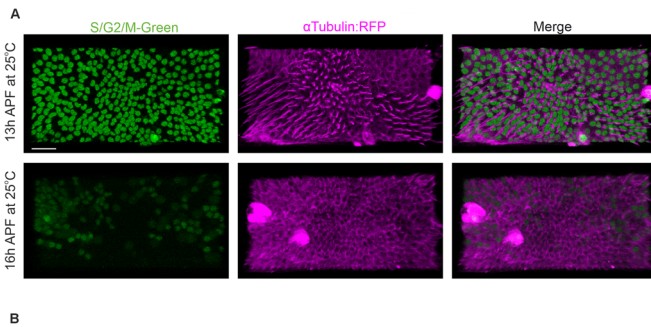

**B**

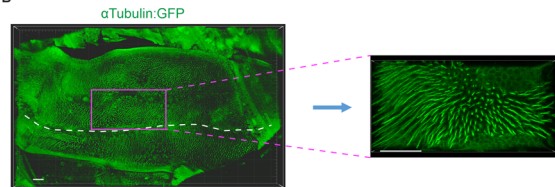

**Figure EV2. Time-lapse imaging of cell cycle changes using S/G/M-green in pupal wing epithelial cells between 13 and 16 h APF.**

(A) Our observations reveal that the majority of cells remain in the S/G2 phase during the early inflation stage (13 h APF at 25 °C). After 3 h (16 h APF at 25 °C), many cells enter mitosis. (B) Delimiting the region of interest (ROI) in which MT protrusion loss and mitoses are counted in dorsal and ventral epithelia. The ROI (magenta box) is adjacent to the trachea (white dashed line) close to the future hinge of the wing. Scale bar: 20 μm (A) and 30 μm (B).

**Figure EV3. Modulating MT stability affects wing morphogenesis.**

(A) Adult wing in control condition I. (B) Lateral view of αTubulin:GFP of control at 10.5, 12.5, and 14.5 h APF. (C) Adult wing overexpressing hTau in both dorsal and ventral epithelium. (D) Lateral view of αTubulin:GFP during hTau overexpression in both dorsal and ventral layers (*nub-Gal4> hTau*) at 10.5, 12.5, and 14.5 h APF. (E) Lateral view of αTubulin:GFP during hTau overexpression in dorsal layers only (*ap-Gal4> hTau*) at 10.5, 12.5, and 14.5 h APF. Note that dorsal protrusions are thicker than ventral protrusions in hTau overexpression only in dorsal cells, which results in loss of cell–cell contacts (arrowheads). (E') Schematics of the loss of cell–cell contacts in hTau oeverexpression only in dorsal cells. Disassembly of MT projections in ventral cells involves degeneration of the basal integrin-laminin complex, but not in dorsal cells, which is sufficient for leading to the loss of cell–cell contact. (F) Adult wing overexpressing Katanin60 in dorsal epithelium. Scale bars: 250 µm (**A, C, F**), 5 µm (**B, D, E**).

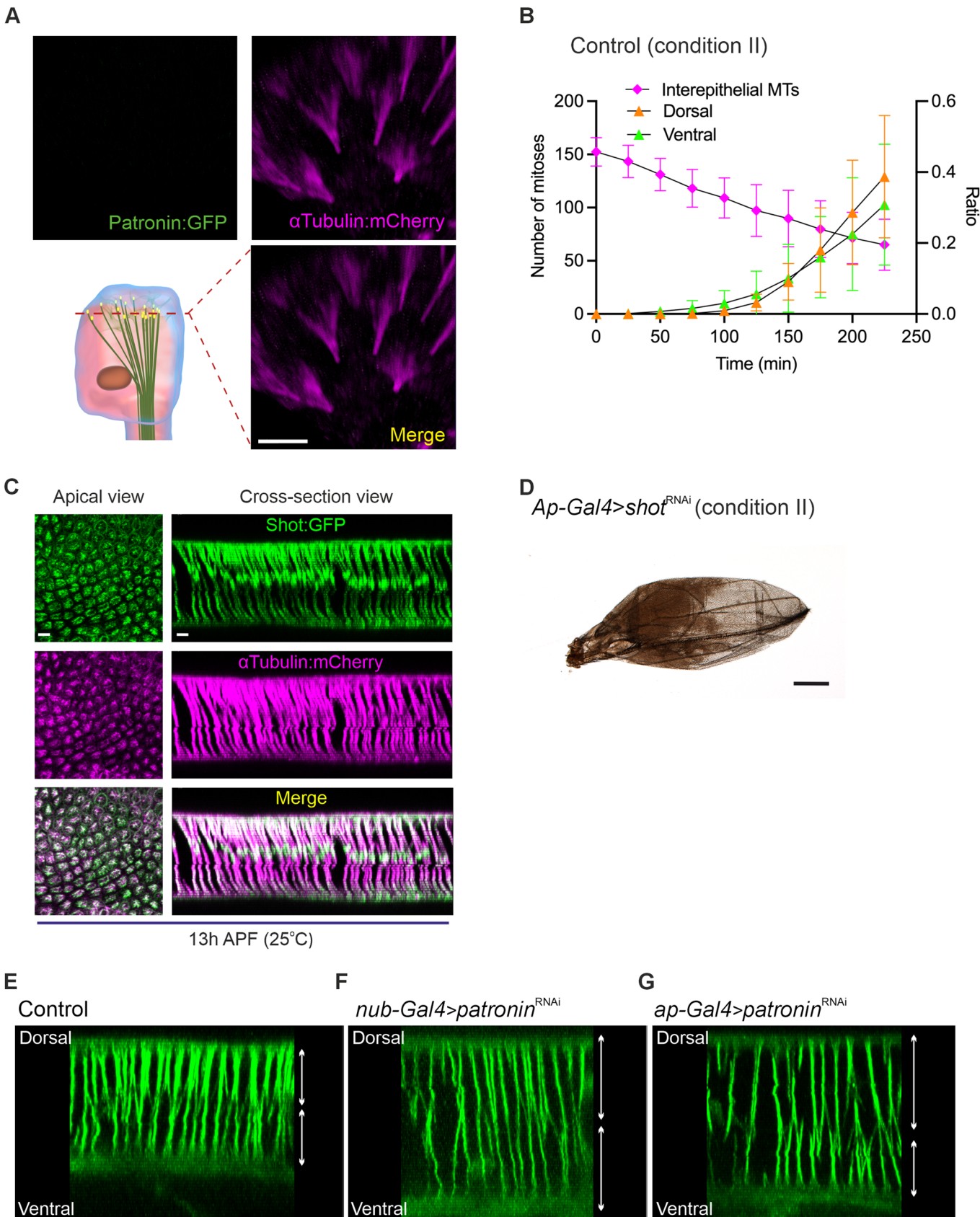

**Figure EV4. MT-based protrusions are sustained by Patronin and Shot.**

(A) Live imaging of αTubulin:mCherry (magenta) and Patronin:GFP (green) at 10.5 h APF (29 °C) during conditional RNAi of *patronin*, Note that Patronin:GFP signal diminishes after RNAi (compare to Fig. 7A). (B) Number of mitotic cells (dorsal: orange triangle, ventral: green triangle) in wing epithelium and ratio of interepithelial MT (magenta) at different time points in control pupal wings (condition II). Time 0 corresponds to 10.5 h APF. Data are from five individual replicates ($N = 5$). Data are means ± 95% CIs. (C) Shot:GFP localization in pupal wings. Live imaging of αTubulin:mCherry and Shot:GFP at 13 h APF (25 °C). Shot.GFP (top), αTubulin:mCherry (middle), merged image (bottom). (D) Adult wing of *shot* RNAi in dorsal epithelium. (E–G) Lateral view of αTubulin:GFP during control (E), conditional *patronin* RNAi in both dorsal and ventral layers (*nub-Gal4> patronin RNAi*, F) or only in dorsal layer (*ap-Gal4> patronin RNAi*, G). Note that protrusions are thinner and longer than control in *patronin^RNAi* in two layered epithelia (E, F), and ventral protrusions are thicker and shorter than dorsal protrusions during only dorsal *patronin^RNAi* (G). Scale bars: 5 µm (A, C), 250 µm (D), 15 µm (E–G).

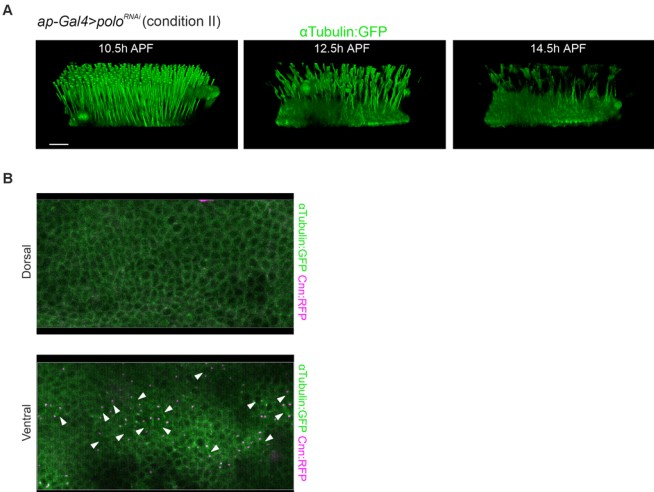

**A** ap-Gal4>polo^RNAi (condition II)    αTubulin:GFP

10.5h APF    12.5h APF    14.5h APF

**B**

Dorsal — αTubulin:GFP Cnn:RFP

Ventral — αTubulin:GFP Cnn:RFP

**Figure EV5. Time lapse images of 3D view of MT protrusions visualized by αTubulin:GFP at 10.5, 12.5, and 14.5 h APF (29 °C) of the wing during conditional knockdown of *polo* (*ap > polo RNAi*).**

(A) Apical surface of the dorsal epithelium is towards the top of the view. (B) Dorsal (top) and ventral (bottom) epithelial cells visualized by αTubulin:GFP (green) and Cnn:RFP (magenta) at 14.5 h APF. Note that mitotic cells are only observed in ventral epithelium (arrowheads). Scale bars: 20 μm (A, B).

