## [Peer Review File · The EMBO Journal]

Programmed disassembly of a microtubule-based membrane protrusion network coordinates 3D epithelial morphogenesis in *Drosophila*

Ngan Tran, Martti Montanari, Jinghua Gui, Dmitri Lubenets, Lea Fischbach, Hanna Antson, Yunxian Huang, Erich Brutus, Yasushi Okada, Yukitaka Ishimoto, Tambet Tõnissoo, and Osamu Shimmi

DOI: [10.15252/emboj.2023113921](https://doi.org/10.15252/emboj.2023113921)

Corresponding author: Osamu Shimmi (osamu.shimmi@helsinki.fi)

Review Timeline:

Submission Date:	1st Mar 23
Editorial Decision:	5th May 23
Revision Received:	6th Nov 23
Editorial Decision:	13th Dec 23
Revision Received:	14th Dec 23
Accepted:	18th Dec 23

Editor: Ieva Gailite

Transaction Report:

Dear Dr. Shimmi,

Thank you for submitting your manuscript for consideration by the EMBO Journal. I apologise for the protracted assessment process due to delays in referee report submission. We have now received comments from three reviewers, which are included below for your information.

As you will see from the reports, all reviewers find the proposed model of cell division coordination between the pupal wing epithelial layers via cytoskeleton-rich protrusions per se interesting. However, they also agree that the mechanism remains currently unclear, and outline a number of experiments that will be needed to substantiate the proposed model. Furthermore, they find the used Interplanar Amida Network terminology unclear, since these protrusions have been as such previously described, and recommend to use the previously published term for these structures (transalar cytoskeletal arrays).

If you find that you are able to address the main issues raised by the reviewers, I would be happy to consider a revised version of the manuscript. I think it would be helpful to discuss the revision in more detail via email or phone/videoconferencing - please let me know which option you prefer. I should also add that it is The EMBO Journal policy to allow only a single major round of revision and that it is therefore important to resolve the main concerns at this stage. If several of the main points cannot be experimentally addressed, the revised manuscript could be potentially suitable for publication in our sister journal EMBO Reports.

We generally allow three months as standard revision time, which can be extended to six months in the case of major revisions. As a matter of policy, competing manuscripts published during this period will not negatively impact on our assessment of the conceptual advance presented by your study. However, please contact me as soon as possible upon publication of any related work to discuss the appropriate course of action. Should you foresee a problem in meeting this deadline, please let us know in advance to discuss an extension.

When preparing your letter of response to the referees' comments, please bear in mind that this will form part of the Review Process File and will therefore be available online to the community. For more details on our Transparent Editorial Process, please visit our website: <https://www.embopress.org/page/journal/14602075/authorguide#transparentprocess>. Please also see the attached instructions for further guidelines on preparation of the revised manuscript.

Please feel free to contact me if you have any further questions regarding the revision. Thank you for the opportunity to consider your work for publication. I look forward to discussing your revision.

With best regards,

Ieva

We realize that it is difficult to revise to a specific deadline. In the interest of protecting the conceptual advance provided by the work, we recommend a revision within 3 months (3rd Aug 2023). Please discuss the revision progress ahead of this time with the editor if you require more time to complete the revisions.

Referee #1:

The authors analyse a complex structure during pupal wing morphogenesis in *Drosophila* they term the IPAN, though it was previously already described as the transalar cytoskeletal arrays. The authors use high resolution and live imaging to study the dynamic behaviour of the IPAN during wing apposition, separation and re-apposition. They uncover that contact between dorsal and ventral epithelial layers is required cell-autonomously and non-autonomously to coordinate mitoses between the wing layers. The processes are built by prominent microtubule bundles that interestingly undergo a change from non-centrosomal to centrosomal arrangement.

One minor comment here, the term IPAN should be explained earlier because it is not clear where the term comes from until the reader has reached the results section.

This study is based on beautiful live imaging in a challenging system and quantitative analysis of wild-type and perturbed situations. This allows the authors to firstly capture in novel details how the two layers of the wing blade interact through basal microtubule protrusions during the first apposition of the wing layers, and how the arrangement of microtubules in basal protrusions relates to commencement of mitoses.

They then go on to test the hypothesis that protrusion presence/disassembly of microtubules and start of mitoses are functionally linked.

Proteins that in other contexts have been shown to coordinate the organisation of microtubules into non-centrosomal arrays, Patronin and Shot, are also important here to maintain the microtubule-based basal protrusion. They go on to show that alpha-Spectrin, that can also interact with Shot and Patronin, affect the mitoses of both epithelial layers as well, through regulation of cell shape, as reduction of alpha-Spectrin leads to changes in the length of different stage of IPAN assembly/presence/disassembly.

The imaging data are beautiful and highly informative, the realisation that there is tight coordination between the layers, even during the stage of basal protrusion disassembly, leading to non-autonomous effects even if microtubules are modified only within one epithelial layer is very intriguing.

I am just still not completely clear on how the authors imagine the mechanism or communication to work. They test forced disassembly of the IPAN by overexpressing Katanin, even of only in the dorsal layer, where it clearly disrupt the IPAN as well as mitoses, but mitosis inhibition also happens on the ventral side, even though these cells have some protrusion (but they don't look wild-type).

Did the authors try to overexpress tau to stabilise MTs only on one side? Does that prevent mitoses on both sides? Or des the normal side still manage to disassemble the IPAN?

Furthermore, how alpha-Spectrin links into all of this is not really clear to me. Is there a feedback from alpha-Spectrin/apical cell shape back onto the microtubules? Do they look or behave different in any way apart from the changes in timing? This should probably be analysed.

There is also a mis-label in Figure 6: panel A states 'nu-Gal4> shotRNAi' when this should now be 'alpha-Spectrin-RNAi'!

Similar to microtubules in the alpha-Spectrin-RNAi, the authors should also show if there is a concomitant change to the myosin intensity and arrangement that fits with the changes in cell shape they describe.

Overall, the microtubule and cell shape part need to be better integrated with one another, and the reasoning for the overall mechanism linking the two need to be better explained and elaborated.

Referee #2:

In this manuscript, Tran et al. provide insights into a previously uncharacterized morphogenetic role of the microtubule-based membrane protrusion network, named the Interplanar Amida Network (IPAN) that dynamically connects the dorsal and ventral epithelium of the emerging *Drosophila* wing. This work presents an important improvement in the non-invasive live imaging method, recording pupal wing development. Presentation of spectacular high-resolution live images of IPAN organization and disassembly during different stages of morphogenesis supports the developmental programming of this protrusion-based network in precise coordination with the regulation of cell division that modulate wing growth. Moreover, the authors showed that the microtubule cores of IPAN originate from non-centrosomal MTOC, and the reverse correlation of the IPAN organization with centrosomal MTOC organization and cell division, suggesting an interesting model where the IPAN assembly and disassembly might control cell division across the emerging wing epithelium. This work is significant, and it provides new insights into how long-distance communication/interactions via various forms of membrane protrusions might coordinate cell shape change, cell division, and dynamics at a global scale.

I have several comments that might improve the work:

1) One major take-home message of the paper is the effects of IPAN connection disassembly, coordinating cell division re-entry in the emerging wing. However, cell division is measured based on only *cnn*:RFP. An alternative marker for cell division (e.g., Fucci/BrdU/p-Histone), if genetically/experimentally possible, showing cell division arrest and re-entry might strengthen the model.

2) Some of the words used in the result section need more clarification:

a) Microfilament (MF) - is this referring to actin microfilaments?

b) IPAN is apparently the same structure as the Transalar cytoskeletal array (TCA) known earlier.

Although, in my opinion, IPAN might indicate a clear picture of the ladder-like network, a justification for the use of a different name for the same architecture might be important to discuss, avoiding future confusion with the nomenclature.

3) Fig.1: An additional schematic (can be integrated with panel A/B), showing - a) the morphogenetic folding of the disc epithelium and the apposition of the dorsal and ventral layers (as described in the Introduction/Results); and (b) the apical and basal faces relative to the objective and/or the imaging planes would be easier to follow the description of 3D morphogenetic events, to better understand the organization of IPAN network presented in actual confocal images. Some of these schematics might be there in supplement figures/videos, but a main Figure panel with the detailed morphological features of the wing development would be helpful to readers from other fields.

4) The following sentence might be difficult to picture: "We found that apical tips of MTs are medially distributed around 13 - 14h APF (Fig 1D)". To my understanding, MT polarity is described as + and - ends. So, what are the apical tips of MTs? And, medial positioning relative to what? Similarly, it would be helpful to explain some of the terms like 'apical view' and 'apical compartment', 'MT spike', and other terminology used to describe IPAN. Probably a schematic would be helpful to define these terms describing relative positioning.

MT foci are visualized in the apical view, but do they originate exactly at the apical cell membrane? Why is the apical membrane described as a compartment? Do ncMTOCs localize only in the apical membrane? Secondly, the images in Fig 1 indicate that the IPAN MTs occupy cell cytoplasm. Does IPAN affect nuclear positioning/shapes, which, in turn, might control cell division? A nuclear stain/cell division marker like Fucci might reveal interesting information.

5) Does an "MT-spike" represent a conical projection composed of 13 MTs or just one MT within a conical projection? From Lifeact:Ruby images, it is apparent that individual MTs are not covered by actin, but the cone-like membrane protrusion that is composed of several MTs has an actin cortex. Rewording the result description of Figure 1 might help. Apparently, differences between actin-binding lifeact:ruby (f-actin) and mCherryCAAX (membrane) localization are hard to notice in the images. Is it simply due to the localization of life-act-stained f-actin in the cell membrane cortex?

6) Fig. 2 (also Movies 2 and 3): Spectacular high-resolution live imaging showing dynamics of dorsal and ventral projections that constitute IPAN across two layers. Please indicate what the colors in Movie 2 represent (probably for wing veins?).

Secondly, I have a suggestion for the following sentence: " suggesting that cell-cell contacts between the two epithelia are maintained through membrane-membrane attachments rather than synapse-like structures."

The context of synapse-like structures might be relevant but might be confusing in the context of IPAN. Synapses are formed by dynamic interactions of pre- and post-synaptic projections/filopodia, and synapses can be dynamic/plastic. Similar to IPAN and Synapses, inter-organ/-cellular cytoneme::cytoneme contacts also dynamically assemble and disassemble to exchange signals like FGF, Hh, and Wnt. So, there is a common/conserved rule of projection-mediated long-distance intercellular interactions in different contexts, but similarity and dissimilarity with synapse might be a speculation without a thorough understanding of these events in IPAN.

Finally, the result description/conclusion in the text provides a picture of uniform dynamic disassembly of IPAN across wing discs. However, based on Movie EV3, the disassembly of projections appears to be localized/zonal, taking place only in regions that form large interlayer gaps (probably forming wing veins). In contrast, projections from two layers are reinforced by bundling and strong adhesion followed by apico-basal shortening of the projections in the intervein regions, thus inwardly pulling two layers to confine each gap/vein region within the emerging wing blade. In these regions, attachments between protrusions could be very strong and stable, involving many synapse-forming molecules (cell adhesion molecules and matrix previously described for TCA). The results can be re-evaluated and commented from this perspective.

7) Fig. 4. Results show that the disassembly/reorganization of the longitudinal MT network is coordinated with the formation of mitotic spindles. This is consistent with the proposed model that the common α -tub-GFP pool is repurposed to form centrosome-spindles to induce cell division. Please provide the quantitation - e.g., no. of cells observed to form mitotic spindles. Secondly, it would be interesting to know if there is any zonal difference in cell division. For instance, what happens to the intervein regions where strong adhesion among inter-layer projections causes bundling and shortening (as suggested in movie 3)? Do these cells also divide at a similar rate?

Another concern on the detection of *cnn*:RFP is that the images were apparently captured only at the selected sections in the Dorsal and Ventral cell body/projection (panel B). Unless the centrosome is known to be present always at the selected optical planes, the quantitation of mitosis based on *cnn* localization might be misleading. As indicated before, an alternative method to detect cell division might be needed.

Secondly, based on Fig 4C-C', 10.5 panel has more *cnn*:RFP puncta (almost everywhere) along with more projections, unlike in the graph in C' indicating low *cnn*. From the figure panels, an alternative interpretation could be that the *cnn*:RFP was uniformly localized in various places at 10.5 and then it got limited to only some regions where projections are lost (during the 12.5 and 14.5 APF). A clarification of these observations might be necessary.

Some sentences in this section, especially on the role of IPAN disassembly in cell division, needs more clarification. Based on the results, stabilization of MT suppressed mitosis, but destabilization of MT did not increase mitosis. This might suggest that the MT disassembly is not the sole regulator of the mitotic re-entry of cells, unlike what stated. Please clarify/discuss.

8) Fig 5. Very clear evidence showing the roles of the *shot* gene in MT organization. Secondly, experiments with *shot* knockdown selectively in one of the two interacting layers clearly suggest IPAN-mediated non-autonomous interactions somehow coordinate growth in two opposing wing layers. However, some of the description/conclusions need more clarification. The alternative possibilities should be verified and discussed.

First, the loss of cell division due to *shot*-RNAi also could be due to the *Shot*'s role in spindle assembly. This possibility was not discussed/verified.

Second, since experiments presented here did not track IPAN assembly under the *shot* knockdown condition, it is difficult to conclude whether programmed disassembly is the only cause of the *shot*-RNAi phenotype. Under the RNAi condition, IPAN network might not have assembled in the first place. So, the interpretation suggesting the disassembly of IPAN and its effects is erroneous in this context.

8) The role of α -spectrin in IPAN-mediated mitosis in Fig 6, is a bit confusing and is thoroughly examined. I am unsure if this section on α -spectrin is presenting any significant information needed for the main message of the paper. This is a personal opinion. There are several correction and reclarification needed:

First, Fig. 6A figure panel caption '*shot* RNAi' might be spectrin RNAi.

Please provide the significance value for the increase in mitosis in α -Spectrin knockdown relative to control. Probably the control and RNAi graph side by side might be easier to compare.

Here, I suppose cellular mechanisms meant mitosis.

Since α -spectrin knockdown does not significantly affect dynamics of IPAN disassembly relative to control but somehow increases mitotic cell numbers, it is unclear how α -Spectrin might regulate IPAN-mediated cellular mechanisms, as suggested in

this section. And the results apparently suggest that spectrin might be required to suppress mitosis independent of IPAN assembly-disassembly. A clarification would be helpful.

"When α -Spectrin was conditionally knocked down, apical compartments showed larger size than control in stage I, and do not show significant expansion from stage I to II" The conclusion is apparently based on only 12 cells in Fig 6B. Please provide quantitative data and actual images (at least in the supplement). Also, in D, cell sizes in stages I and II appear to be the same in control, unlike those shown in C and B. Please explain more about why there are inconsistencies in the size of cells.

Cortical Sqh level is reduced in stage II in Fig 6D; is this due to the relaxation of cells? The relationship of Sqh localization to different parts of cells with cell constriction and relaxation could be clarified more for general readers.

Referee #3:

Summary of findings

In this manuscript, the authors employ non-invasive live imaging to investigate the mechanisms of *Drosophila* wing development during the pupal stage, when two epithelial layers (dorsal and ventral) undergo initial apposition, separation, and re-apposition in a coordinated manner.

The authors show that coordination of growth and mitosis in two spatially separated wing epithelial layers requires formation of a cytoskeletal network, consisting of long basal cell projections from both the dorsal and ventral cells. This dynamic structure, formed by microtubules and microfilaments, is termed by the authors as the Interplanar Amida Network (IPAN). Vertical bundles are further interconnected by lateral actin filaments. The authors show that following formation during wing separation stage, the IPAN is progressively disassembled prior to mitosis in both epithelial layers. Stabilisation of microtubules, or their ectopic disassembly in the dorsal region reduced mitosis in both epithelia, indicative of cell non-autonomous roles for the IPAN. The microtubule component of projections is organised by non-centrosomal microtubule organising centre (ncMTOC): by depleting the ncMTOC components, Patronin and Shot, formation of the IPAN is abolished. Furthermore, cell shape dynamics mediated by α -Spectrin were shown to regulate IPAN assembly and disassembly.

By employing a technically challenging microdissection protocol for non-invasive live imaging, detailed visualisation of IPAN dynamics enabling quantification was achieved by the authors. This evidence for coordination of tissue growth by the IPAN represents a novel, and significant, observation.

The mechanisms of IPAN organisation reported will be of broad interest to developmental biologists investigating principles of 3D tissue morphogenesis. Nevertheless, the novelty behind the IPAN, and specifically the need for new terminology, was not clear as these structures were described as early as 1940. Specifically, as the structure was previously named the Transalar Cytoskeletal Array (TCA), as the authors themselves noted, is it necessary to rename this structure?

Additional concerns with methodology and points requiring clarification by the authors' attention are detailed below.

Major comments

(1) Although the authors coin the term IPAN, it is unclear why the new name is necessary as the structures have been previously referred to as transalar cytoskeletal arrays (TCAs) in the literature. The structures have also been observed in early studies (although their dynamic nature was only appreciated recently, e.g. by Sun et al 2021 <https://doi.org/10.1016/j.celrep.2021.109667>). To warrant the proposed new nomenclature, more discussion of specific novelty of the IPAN and its distinction from TCA is required.

(2) Re-apposition of epithelia is described to be mediated by cytoskeletal features, but this was only shown in terms of membrane structure (CAAX:mCherry) at >15h APF (Figure 2 A, B), yet images of tubulin:GFP are not shown at this time point. The authors should include images of tubulin:GFP at 15-24h APF to support this claim.

(3) The wing expression studies using *tsgal80* were performed at room temperature before shifting to 29 degrees. *Tsgal80* is active at 18, partially active at 25 and inactive at 29 (McGuire et al 2004 10.1126/stke.2202004pl6). It is, therefore, possible GAL4 expression was active at earlier time points.

(4) It is critical for the authors to validate the effectiveness of knockdown for the lines in this study at both temperatures, and perform genetic crosses at 18 before shifting to 29.

(5) Given the potential for off-target effects, it is also imperative to demonstrate reproducibility with alternate RNAi lines, e.g. Patronin RNAi #108927 used in this study has 1 predicted off-target https://shop.vbc.ac.at/vdrc_store/108927.html.

(6) The authors show that destabilising MTs in the dorsal compartment, by expressing *Kat60*, non-autonomously disrupts

mitoses in the ventral compartment. This is proposed to be indicative of cell-cell contacts being required for mitoses. Although it is possible that the disassembly process itself is required to initiate mitosis, as the IPAN is never formed in this condition, the data are not sufficient to support this conclusion. To confirm such non-autonomous roles of the IPAN, analysis should also be done following stabilisation of the microtubules specifically in the dorsal compartment using ap-Gal4>hTau.

(7) The authors conclude that knockdown of Patronin leads to incomplete loss of microtubules, but again the efficiency of the RNAi for knockdown is not demonstrated. The authors need to determine whether Patronin mRNA and/or protein is depleted by qPCR or antibody staining and confirm the phenotype with an alternate Patronin RNAi line.

(8) While quantitative studies of mitoses and protrusions are referred to as "significant", this is not always obvious - e.g. from needing to compare a-spectrin RNAi which appears in Figure 4A, to the control which appears in Figure EV4D. It would be helpful to indicate statistical significance on the graphs.

Minor comments

(1) It would be helpful to clarify in the text what lifeAct marks, i.e. actin microfilaments

(2) Although the authors show that cell protrusions include both microtubule and actin components, including lateral connections mediated by filopodia-like structures, the latter investigations focus on tubulin alone, which should be justified in the text.

(3) The images in Figure 3 and EV2 are duplicated - the cnn:RFP panels alone should be shown in the main figure for clarity.

(4) In the main text, clarify the rationale for switching to the 29 degree temperature (i.e. enabling transgene expression, overexpression in condition I and knockdown in condition II)

(5) Although I appreciate that different stages are written in the legends, it would be useful to have the relevant descriptions overwritten on the video EV4 alongside the arrows at 27 seconds, 31 seconds, etc.

(6) There is an incorrect label on Figure 6A - the graph shows a-spectrin rather than shot RNAi?

(7) The size of cells images in Figure 6D is not consistent with apical relaxation - the cells in Stage II appear the same or smaller than stage I.

(8) Known/expected roles of Sqh in apical relaxation should be clarified in the text.

We appreciate the valuable comments from the three referees. Taking all comments into consideration, we decided to revise the manuscript by focusing on two themes.

First, we introduce the dynamics of the IPAN structure in more detail. This involves time-lapse images of lateral filopodia-like structures coupled with MT protrusion dynamics (Fig. 2B, C, Movie EV3). In addition, time-lapse images of α Tubulin:GFP at the tissue scale, as well as time-lapse images of membrane-bound CAAX:mCherry, provide the insights that MT dynamics play a key role in IPAN-mediated 3D tissue architecture formation (Fig. 2A, D, Movies EV2, 4). Moreover, time-lapse images of two-color images reveal that cell-cell contacts between dorsal and ventral layers are largely maintained (Fig. 3C, D, Movie EV5).

Second, we carefully analyzed how disassembly of MT protrusions leads to mitoses. Our new observations reveal that loss of cell-cell contacts seems to be a key process to provide the conditions for releasing G2 arrest. Our data also support that loss of cell-cell contacts can take place both prior to and after bundle formation (Fig. 4C-E). Additionally, our data also support the hypothesis that uncharacterized mechanisms appear to regulate loss of cell-cell contacts through degeneration of integrin-laminin complex at the basal tips of protrusions. Since degeneration of the ECM complex in only a single layer of epithelial cells seems to be sufficient for releasing cell-cell contacts, we can explain why ectopic expression of hTau or RNAi knockdown of patronin in two layers or one layer shows the different quantity of mitotic cells (Figs 6, 7). Finally, G2/M execution takes place autonomously, therefore loss of Cdc25 in one layer disrupts coordinated mitoses. Below we outline our detailed responses to the referees' comments.

Referee #1:

The authors analyse a complex structure during pupal wing morphogenesis in Drosophila they term the IPAN, though it was previously already described as the transalar cytoskeletal arrays. The authors use high resolution and live imaging to study the dynamic behaviour of the IPAN during wing apposition, separation and re-apposition. They uncover that contact between dorsal and ventral epithelial layers is required cell-autonomously and non-autonomously to coordinate mitoses between the wing layers. The processes are built by prominent microtubule bundles that interestingly undergo a change from non-centrosomal to centrosomal arrangement.

One minor comment here, the term IPAN should be explained earlier because it is not clear where the term comes from until the reader has reached the results section.

We appreciate the comment. The explanation of the term IPAN is provided in detail in the introduction.

This study is based on beautiful live imaging in a challenging system and quantitative analysis of wild-type and perturbed situations. This allows the authors to firstly capture in novel details how the two layers of the wing blade interact through basal microtubule protrusions during the first apposition of the wing layers, and how the arrangement of microtubules in basal protrusions relates to commencement of mitoses.

They then go on to test the hypothesis that protrusion presence/disassembly of microtubules and start of mitoses are functionally linked.

Proteins that in other contexts have been shown to coordinate the organisation of microtubules into non-centrosomal arrays, Patronin and Shot, are also important here to maintain the microtubule-based basal protrusion. They go on to show that alpha-Spectrin,

that can also interact with Shot and Patronin, affect the mitoses of both epithelial layers as well, through regulation of cell shape, as reduction of alpha-Spectrin leads to changes in the length of different stage of IPAN assembly/presence/disassembly.

The imaging data are beautiful and highly informative, the realisation that there is tight coordination between the layers, even during the stage of basal protrusion disassembly, leading to non-autonomous effects even if microtubules are modified only within one epithelial layer is very intriguing.

I am just still not completely clear on how the authors imagine the mechanism or communication to work. They test forced disassembly of the IPAN by overexpressing Katanin, even of only in the dorsal layer, where it clearly disrupt the IPAN as well as mitoses, but mitosis inhibition also happens on the ventral side, even though these cells have some protrusion (but they don't look wild-type). Did the authors try to overexpress tau to stabilise MTs only on one side? Does that prevent mitoses on both sides? Or des the normal side still manage to disassemble the IPAN?

Since overexpression of Katanin60 only in dorsal cells disrupts dorsal MT protrusions, no cell-cell contact between the two epithelia was observed even in the early inflation stage (Fig. 6C). Thus, we consider that initial cell-cell contact is a prerequisite of G2 arrest release through IPAN dynamics.

As suggested by the Referee, we tested overexpression of hTau only in the dorsal layer (*ap-Gal4>hTau*). We found that there were no significant differences in mitoses between the control and *ap-Gal4>hTau* (Fig. 6B). Although hTau overexpression seems to lead to more robust MT protrusions in dorsal cells (Fig. EV3B, D, E), our proposed model suggests that uncharacterized mechanisms regulate loss of cell-cell contacts through degeneration of ECM at the basal tips of protrusions. Since degeneration of ECM in only a single layer of epithelial cells seems to be sufficient for releasing cell-cell contacts, we interpret the data to explain why ectopic expression of hTau in two layers or one layer shows different mitotic phenotypes.

Furthermore, how alpha-Spectrin links into all of this is not really clear to me. Is there a feedback from alpha-Spectrin/apical cell shape back onto the microtubules? Do they look or behave different in any way apart from the changes in timing? This should probably be analysed.

In the revised manuscript, we focus on coordinated mitotic events between the two-layered epithelia. Since α -Spectrin itself affects mitosis only in a very limited manner, data related to α -Spectrin are not included. We appreciate the referee's comments and will address the potential roles for α -Spectrin in future studies.

*There is also a mis-label in Figure 6: panel A states '*nu-Gal4> shotRNAi*' when this should now be '*alpha-Spectrin-RNAi*'!*

This does not apply to the revised manuscript.

Similar to microtubules in the alpha-Spectrin-RNAi, the authors should also show if there is a concomitant change to the myosin intensity and arrangement that fits with the changes in cell shape they describe.

This does not apply to the revised manuscript.

Overall, the microtubule and cell shape part need to be better integrated with one another, and the reasoning for the overall mechanism linking the two need to be better explained and elaborated.

Although we do not include cell shape change data focused on the apical compartment in the revised manuscript, our new data show that MT protrusion dynamics and MF based filopodia-like structure dynamics provide novel data for how MTs and MFs interact in a dynamic manner to support cell shape changes, which support the coupling of growth and patterning of 3D tissue (summarized in Fig. 9).

Referee #2:

In this manuscript, Tran et al. provide insights into a previously uncharacterized morphogenetic role of the microtubule-based membrane protrusion network, named the Interplanar Amida Network (IPAN) that dynamically connects the dorsal and ventral epithelium of the emerging Drosophila wing. This work presents an important improvement in the non-invasive live imaging method, recording pupal wing development. Presentation of spectacular high-resolution live images of IPAN organization and disassembly during different stages of morphogenesis supports the developmental programming of this protrusion-based network in precise coordination with the regulation of cell division that modulate wing growth. Moreover, the authors showed that the microtubule cores of IPAN originate from non-centrosomal MTOC, and the reverse correlation of the IPAN organization with centrosomal MTOC organization and cell division, suggesting an interesting model where the IPAN assembly and disassembly might control cell division across the emerging wing epithelium. This work is significant, and it provides new insights into how long-distance communication/interactions via various forms of membrane protrusions might coordinate cell shape change, cell division, and dynamics at a global scale.

I have several comments that might improve the work:

*One major take-home message of the paper is the effects of IPAN connection disassembly, coordinating cell division re-entry in the emerging wing. However, cell division is measured based on only *cnn*:RFP. An alternative marker for cell division (e.g., Fucci/BrdU/p-Histone), if genetically/experimentally possible, showing cell division arrest and re-entry might strengthen the model.*

To measure cell division, we used both Cnn:RFP and spindle formation visualized by α -Tubulin:GFP (Fig. 4A). To further confirm that cell division takes place, we used the S/G2/M-Green system, which revealed that the majority of the cells during the early inflation stage remain in the S/G2 stage, and that M phase cells start to be observed thereafter (Fig. EV2A), consistent with our Cnn:RFP and α -Tubulin:GFP data. Therefore, we conclude that our current protocol using Cnn:RFP and α -Tubulin:GFP to track cell division is robust.

Some of the words used in the result section need more clarification:

a) Microfilament (MF) - is this referring to actin microfilaments?

MF indeed refers to actin microfilaments.

IPAN is apparently the same structure as the Transalar cytoskeletal array (TCA) known earlier. Although, in my opinion, IPAN might indicate a clear picture of the ladder-like

network, a justification for the use of a different name for the same architecture might be important to discuss, avoiding future confusion with the nomenclature.

We appreciate Referee's comments that the term "IPAN" indicates a clear description of the images we have observed. The following observations further support the notion that IPAN is a more appropriate term than the Transalar cytoskeletal array (TCA):

- 1) Our detailed analysis reveals that the structure not only comprises the vertical protrusions, but also lateral filopodia-like structures between the vertical protrusions (Fig. 1). Thus, the structure is more complex than the previous term "TCA" would suggest.
- 2) Our time-lapse imaging on a tissue scale suggests that the observed cellular structures of the IPAN are not stable, but rather transient, followed by disassembly of some protrusions and bundling of others (Fig. 2, Movie EV2). The term "TCA" insufficiently captures this transient dynamic architecture.
- 3) Our latest time-lapse images focused on lateral filopodia-like structure dynamics further suggest that dynamic lateral structures appear to play critical roles in the dynamics of protrusion structures through facilitating disassembly and bundling of the vertical protrusions (Fig. 2, Movie EV3).

The vertical protrusions, together with the lateral filopodia-like structures, are reminiscent of ladders that evoke *amidakuji*, the Japanese ladder game, thus inspiring the "amida" part of the IPAN acronym. Thus, we consider that IPAN is a more suitable term to represent the structure and its function.

Fig.1: An additional schematic (can be integrated with panel A/B), showing - a) the morphogenetic folding of the disc epithelium and the apposition of the dorsal and ventral layers (as described in the Introduction/Results); and (b) the apical and basal faces relative to the objective and/or the imaging planes would be easier to follow the description of 3D morphogenetic events, to better understand the organization of IPAN network presented in actual confocal images. Some of these schematics might be there in supplement figures/videos, but a main Figure panel with the detailed morphological features of the wing development would be helpful to readers from other fields.

We appreciate the comment. Accordingly, new schematics were added in the Fig. 1A.

The following sentence might be difficult to picture: "We found that apical tips of MTs are medially distributed around 13 - 14h APF (Fig 1D)". To my understanding, MT polarity is described as + and - ends. So, what are the apical tips of MTs? And, medial positioning relative to what? Similarly, it would be helpful to explain some of the terms like 'apical view' and 'apical compartment', 'MT spike', and other terminology used to describe IPAN. Probably a schematic would be helpful to define these terms describing relative positioning.

We appreciate the comments, and have changed the wording accordingly. We added schematics to represent the 3D view of cell shape and MT projections (Fig. 1F). We also clarified the positioning of the apical tips of MTs at different time points using schematics (Fig. 4B). Following the reviewer's suggestions, we illustrated the terms, 'apical view', in Fig. 1. To avoid confusion, we did not use 'apical compartment' and 'MT spike' in the revised manuscript.

MT foci are visualized in the apical view, but do they originate exactly at the apical cell membrane? Why is the apical membrane described as a compartment? Do ncMTOCs localize

only in the apical membrane? Secondly, the images in Fig 1 indicate that the IPAN MTs occupy cell cytoplasm. Does IPAN affect nuclear positioning/shapes, which, in turn, might control cell division? A nuclear stain/cell division marker like FUCCI might reveal interesting information.

Our high-resolution imaging of α -Tubulin:GFP and LifeAct:Ruby reveals that the minus-ends of MT protrusions are co-localized with apical microfilament structures, indicating that they originate from the apical part of the cells.

The term “apical compartment” is generally used for the part of the epithelial cells that face the lumen or the external environment, containing special structures and functions such as absorption and secretion. The apical part of the pupal wing epithelial cells has been insufficiently explored, thus structural and functional details are not well known. To avoid misunderstanding, we have replaced the term “apical compartment” with “the apical region”. In Fig. 7, we investigated the distribution of Patronin, a key component of ncMTOCs. Our observations clearly indicate that Patronin is specifically localized at the apical tips of MT protrusions. Therefore, we conclude that Patronin-containing ncMTOCs are localized apically.

To understand the relationship of the IPAN to nuclear position/shape, we employed imaging using α Tubulin:RFP and nuclear membrane-bound GFP. As shown in the revised Fig. EV1D, MTs forming the IPAN localize away from the nucleus, and the cytoplasm of the apical half of the cells is largely occupied by MT protrusions, thus the position of the nucleus appears to be restricted during this stage.

Does an "MT-spike" represent a conical projection composed of 13 MTs or just one MT within a conical projection? From Lifeact:Ruby images, it is apparent that individual MTs are not covered by actin, but the cone-like membrane protrusion that is composed of several MTs has an actin cortex. Rewording the result description of Figure 1 might help. Apparently, differences between actin-binding lifeact:ruby (f-actin) and mCherryCAAX (membrane) localization are hard to notice in the iamges. Is it simply due to the localization of life-act-stained f-actin in the cell membrane cortex?

An MT-spike represents a conical projection composed of multiple arrays of MTs. Following the reviewer’s recommendation, we have revised the description of microfilaments forming membrane protrusions. We acknowledge that the majority of F-actin is utilized for membrane protrusions rather than for surrounding individual MT projections. As a result, the F-actin image closely resembles the cell membrane cortex.

Fig. 2 (also Movies 2 and 3): Spectacular high-resolution live imaging showing dynamics of dorsal and ventral projections that constitute IPAN across two layers. Please indicate what the colors in Movie 2 represent (probably for wing veins?).

We have added a detailed explanation in Movie EV2.

Secondly, I have a suggestion for the following sentence: " suggesting that cell-cell contacts between the two epithelia are maintained through membrane-membrane attachments rather than synapse-like structures."

The context of synapse-like structures might be relevant but might be confusing in the context of IPAN. Synapses are formed by dynamic interactions of pre-and post-synaptic projections/filopodia, and synapses can be dynamic/plastic. Similar to IPAN and Synapses, inter-organ/-cellular cytoneme::cytoneme contacts also dynamically assemble and

disassemble to exchange signals like FGF, Hh, and Wnt. So, there is a common/conserved rule of projection-mediated long-distance intercellular interactions in different contexts, but similarity and dissimilarity with synapse might be a speculation without a thorough understanding of these events in IPAN.

We appreciate the referee's feedback. We concur that the usage of the term 'synapse-like structures' might be confusing. In the revised manuscript, we clarify that these interactions occur at the basal membranes via ECM.

Finally, the result description/conclusion in the text provides a picture of uniform dynamic disassembly of IPAN across wing discs. However, based on Movie EV3, the disassembly of projections appears to be localized/zonal, taking place only in regions that form large interlayer gaps (probably forming wing veins). In contrast, projections from two layers are reinforced by bundling and strong adhesion followed by apico-basal shortening of the projections in the intervein regions, thus inwardly pulling two layers to confine each gap/vein region within the emerging wing blade. In these regions, attachments between protrusions could be very strong and stable, involving many synapse-forming molecules (cell adhesion molecules and matrix previously described for TCA). The results can be re-evaluated and commented from this perspective.

We appreciate the insightful comments. We provide wider-area imaging of α Tubulin:GFP to demonstrate how disassembly and bundling of MT protrusions occur at the tissue level (Fig. 2D, Movie EV4) in addition to membrane bound CAAX-mCherry images (Fig. 2A, Movie EV2). Moreover, our new data include dynamics of both α Tubulin:GFP and CAAX-mCherry in zoomed-in images to interpret how vertical MT protrusions and lateral filopodia-like structures interact to support disassembly and bundling of the IPAN (Fig. 2B, C, Movie EV3).

Our revised data reveal the following. First, disassembly and bundling of MT protrusions take place throughout the wing tissue, even though the timing differs in different positions (Fig. 2A, D, Movies, EV2, 4). Since capturing mitotic cells require a zoomed-in image, it is only possible for us to monitor mitotic cells in a limited area (Figs 4, 5, Fig. EV2B).

Nonetheless, our data sufficiently support that IPAN dynamics via MT disassembly contribute to both coordinated mitoses (growth) and wing morphogenesis (patterning), as summarized in Fig. 9.

Second, cellular behavior between future intervein/hinge and vein cells appears different. Since mitotic cells are also observed in the future vein cells without the IPAN structure, we hypothesize that different mechanisms are involved in regulating mitosis for them. We plan to investigate detailed molecular mechanisms in future intervein/hinge (with MT protrusions) and future vein (without MT protrusions) as part of our ongoing plans.

Fig. 4. Results show that the disassembly/reorganization of the longitudinal MT network is coordinated with the formation of mitotic spindles. This is consistent with the proposed model that the common α -tub-GFP pool is repurposed to form centrosome-spindles to induce cell division. Please provide the quantitation - e.g., no. of cells observed to form mitotic spindles.

Our quantification protocol includes manual input to count the number of MT protrusions and the number of centrosomes (Cnn:RFP) / spindles formed (α -Tubulin:GFP). These results support the idea that Tubulin is repurposed from MT protrusions to mitotic spindle formation. In addition, the high resolution time-lapse imaging of α -Tubulin:GFP reveals that the mitotic spindle is formed after the degeneration of MT protrusions (Fig. 4A). Therefore, our

quantification protocol precisely represents the number of cells forming mitotic spindles as mitotic cells.

Secondly, it would be interesting to know if there is any zonal difference in cell division. For instance, what happens to the intervein regions where strong adhesion among inter-layer projections causes bundling and shortening (as suggested in movie 3)? Do these cells also divide at a similar rate?

As shown below, we have observed differences in cell division when we marked mitotic cells between 10.5-14.5h APF (29°C) only in dorsal cells. We assume that different dynamics of the IPAN and other molecules, such as growth factor signaling, regulate this. Since the interepithelial distance is largest at the distal tip of the wing ($\geq 150 \mu\text{m}$), our current protocol is not able to capture adequately images of ventral cells. Considering that the main focus of this article is to address cell-cell communication between the two epithelia during 3D morphogenesis, we do not include such data in our current manuscript. We are developing a new protocol to capture a wider 3D area of 3D tissue by using another type of microscopy, and we hope we can address the question of zonal differences in 3D tissue in our future research.

α -Tubulin:GFP (green) and mitotic cells during 10.5-14.5h APF (magenta).

*Another concern on the detection of *cnn*:RFP is that the images were apparently captured only at the selected sections in the Dorsal and Ventral cell body/projection (panel B). Unless the centrosome is known to be present always at the selected optical planes, the quantitation of mitosis based on *cnn* localization might be misleading. As indicated before, an alternative method to detect cell division might be needed.*

As discussed above, our protocol includes not only counting Cnn:RFP, but also mitotic spindle formation using α -Tubulin:GFP. Quantification is not derived from a single plane, instead we combined planes covering cell bodies. Thus, we ensure that our protocol has a minimal risk of including false positives/negatives.

*Secondly, based on Fig 4C-C', 10.5 panel has more *cnn*:RFP puncta (almost everywhere) along with more projections, unlike in the graph in C' indicating low *cnn*. From the figure panels, an alternative interpretation could be that the *cnn*:RFP was uniformly localized in various places at 10.5 and then it got limited to only some regions where projections are lost (during the 12.5 and 14.5 APF). A clarification of these observations might be necessary.*

Regarding the Cnn:RFP puncta of the 10.5h APF panel, we believe that these puncta represent aggregated proteins rather than being part of the pericentriolar material complex,

and therefore, they do not indicate mitotic cells. As we mentioned before, our approach for quantifying mitotic cells considers not only Cnn:RFP puncta, but also spindle formation observed in α Tubulin:GFP images. This methodology helps prevent the inclusion of non-mitotic Cnn:RFP puncta for quantification analysis.

Some sentences in this section, especially on the role of IPAN disassembly in cell division, needs more clarification. Based on the results, stabilization of MT suppressed mitosis, but destabilization of MT did not increase mitosis. This might suggest that the MT disassembly is not the sole regulator of the mitotic re-entry of cells, unlike what stated. Please clarify/discuss.

We appreciate the insightful comments. In our revised manuscript, we propose that loss of cell-cell contacts, but not MT disassembly itself, plays a key role in regulating the release of G2 arrest. Our new data show that cells overexpressing hTau only in dorsal epithelium (MT stabilized in dorsal protrusions) undergo mitosis, which can be interpreted as loss of cell-cell contacts being mediated via ECM degeneration of basal tips of ventral protrusions (Fig. 6B). Our proposed model is summarized in Fig. 9. We hope this clearly explains our hypothesis.

Fig 5. Very clear evidence showing the roles of the shot gene in MT organization. Secondly, experiments with shot knockdown selectively in one of the two interacting layers clearly suggest IPAN-mediated non-autonomous interactions somehow coordinate growth in two opposing wing layers. However, some of the description/conclusions need more clarification. The alternative possibilities should be verified and discussed. First, the loss of cell division due to shot-RNAi also could be due to the Shot's role in spindle assembly. This possibility was not discussed/verified.

In our revised manuscript, we used the dorsal specific driver *ap-Gal4* to drive *shot^{RNAi}* (Fig. 7E). Although the ventral cells have wild-type genotypes, our observations reveal that mitoses in ventral cells are largely abrogated due to the loss of Shot in the dorsal cells. These data not only show a non-autonomous mechanism, but also the dominant function of Shot in IPAN-mediated morphogenesis. Therefore, our data did not show a direct role for Shot in spindle assembly.

Second, since experiments presented here did not track IPAN assembly under the shot knockdown condition, it is difficult to conclude whether programmed disassembly is the only cause of the shot-RNAi phenotype. Under the RNAi condition, IPAN network might not have assembled in the first place. So, the interpretation suggesting the disassembly of IPAN and its effects is erroneous in this context.

We have captured the IPAN structure as early as possible, which allows us to visualize the image at the early inflation stage. Our data consistently reveal the 'no cell-cell contact phenotype' (Fig. 7E). Furthermore, overexpression of Katanin 60 only in dorsal cells shows similar phenotypes (no cell-cell contact, very low number of mitotic cells, Fig. 6C). Combining these results enable us to propose our hypothesis that establishing cell-cell contacts is a prerequisite for IPAN-mediated coordinated mitoses (Fig. 9).

The role of α -spectrin in IPAN-mediated mitosis in Fig 6, is a bit confusing and is thoroughly examined. I am unsure if this section on α -spectrin is presenting any significant information needed for the main message of the paper. This is a personal opinion. There are several correction and reclarification needed:

We appreciate the referee's comment. In the revised manuscript, we focus on coordinated mitotic events between the two epithelia. Since α -Spectrin itself affects mitosis only in a very limited manner, data related to α -Spectrin are not included in this manuscript. We appreciate the referee's comments below and will address the roles of α -Spectrin in future studies.

First, Fig. 6A figure panel caption 'shot RNAi' might be spectrin RNAi. Please provide the significance value for the increase in mitosis in α -Spectrin knockdown relative to control. Probably the control and RNAi graph side by side might be easier to compare.

Since α -spectrin knockdown does not significantly affect dynamics of IPAN disassembly relative to control but somehow increases mitotic cell numbers, it is unclear how α -Spectrin might regulate IPAN-mediated cellular mechanisms, as suggested in this section. And the results apparently suggest that spectrin might be required to suppress mitosis independent of IPAN assembly-disassembly. A clarification would be helpful.

"When α -Spectrin was conditionally knocked down, apical compartments showed larger size than control in stage I, and do not show significant expansion from stage I to II" The conclusion is apparently based on only 12 cells in Fig 6B. Please provide quantitative data and actual images (at least in the supplement). Also, in D, cell sizes in stages I and II appear to be the same in control, unlike those shown in C and B. Please explain more about why there are inconsistencies in the size of cells.

Cortical Sqh level is reduced in stage II in Fig 6D; is this due to the relaxation of cells? The relationship of Sqh localization to different parts of cells with cell constriction and relaxation could be clarified more for general readers.

Referee #3:

Summary of findings

In this manuscript, the authors employ non-invasive live imaging to investigate the mechanisms of Drosophila wing development during the pupal stage, when two epithelial layers (dorsal and ventral) undergo initial apposition, separation, and re-apposition in a coordinated manner.

The authors show that coordination of growth and mitosis in two spatially separated wing epithelial layers requires formation of a cytoskeletal network, consisting of long basal cell projections from both the dorsal and ventral cells. This dynamic structure, formed by microtubules and microfilaments, is termed by the authors as the Interplanar Amida Network (IPAN). Vertical bundles are further interconnected by lateral actin filaments. The authors show that following formation during wing separation stage, the IPAN is progressively disassembled prior to mitosis in both epithelial layers. Stabilisation of microtubules, or their ectopic disassembly in the dorsal region reduced mitosis in both epithelia, indicative of cell non-autonomous roles for the IPAN. The microtubule component of projections is organised by non-centrosomal microtubule organising centre (ncMTOC): by depleting the ncMTOC components, Patronin and Shot, formation of the IPAN is abolished. Furthermore, cell shape dynamics mediated by α -Spectrin were shown to regulate IPAN assembly and disassembly. By employing a technically challenging microdissection protocol for non-invasive live imaging, detailed visualisation of IPAN dynamics enabling quantification was achieved by the authors. This evidence for coordination of tissue growth by the IPAN represents a novel, and significant, observation.

The mechanisms of IPAN organisation reported will be of broad interest to developmental biologists investigating principles of 3D tissue morphogenesis. Nevertheless, the novelty behind the IPAN, and specifically the need for new terminology, was not clear as these structures were described as early as 1940. Specifically, as the structure was previously named the Transalar Cytoskeletal Array (TCA), as the authors themselves noted, is it necessary to rename this structure?

(1) Although the authors coin the term IPAN, it is unclear why the new name is necessary as the structures have been previously referred to as transalar cytoskeletal arrays (TCAs) in the literature. The structures have also been observed in early studies (although their dynamic nature was only appreciated recently, e.g. by Sun et al 2021

<https://doi.org/10.1016/j.celrep.2021.109667>). To warrant the proposed new nomenclature, more discussion of specific novelty of the IPAN and its distinction from TCA is required.

We appreciate the Referee's comments that our observations are novel and significant. Concerning the terminology, the following observations further support the notion that IPAN is a more appropriate term than the Transalar cytoskeletal array (TCA):

- 1) Our detailed analysis reveals that the structure not only comprises the vertical protrusions, but also lateral filopodia-like structures between the vertical protrusions (Fig. 1). Thus, the structure is more complex than the previous term "TCA" would suggest.
- 2) Our time-lapse imaging on a tissue scale suggests that the observed cellular structures of the IPAN are not stable, but rather transient, followed by disassembly of some protrusions and bundling of others (Fig. 2, Movie EV2). The term "TCA" insufficiently captures this transient dynamic architecture.
- 3) Our latest time-lapse images focused on lateral filopodia-like structure dynamics further suggest that dynamic lateral structures appear to play critical roles in the dynamics of protrusion structures through facilitating disassembly and bundling of the vertical protrusions (Fig. 2, Movie EV3).

The vertical protrusions, together with the lateral filopodia-like structures, are reminiscent of ladders that evoke *amidakuji*, the Japanese ladder game, thus inspiring the "amida" part of the IPAN acronym. Thus, we consider that IPAN is a more suitable term to represent the structure and its function.

(2) Re-apposition of epithelia is described to be mediated by cytoskeletal features, but this was only shown in terms of membrane structure (CAAX:mCherry) at >15h APF (Figure 2 A, B), yet images of tubulin:GFP are not shown at this time point. The authors should include images of tubulin:GFP at 15-24h APF to support this claim.

We have generated time-lapse imaging of α -Tubulin:GFP at the tissue level to illustrate MT protrusions forming interepithelial cytoskeletal features in the revised manuscript (Fig. 2D, Movie EV4).

*(3) The wing expression studies using *tsgal80* were performed at room temperature before shifting to 29 degrees. *Tsgal80* is active at 18, partially active at 25 and inactive at 29 (McGuire et al 2004 10.1126/stke.2202004p16). It is, therefore, possible GAL4 expression was active at earlier time points.*

(4) It is critical for the authors to validate the effectiveness of knockdown for the lines in this study at both temperatures, and perform genetic crosses at 18 before shifting to 29.

We maintain the fly crosses at room temperature (around 21 – 22°C) before activating the UAS/Gal4 system (by inactivating temperature-sensitive Gal80) at 29°C. To confirm that Gal80^{ts} is sufficiently functional at room temperature, but not at 29°C, we tested GFP expression using the genotype *nub*-Gal4>UAS-GFP, Gal80^{ts} and collected live images under the identical conditions, except for temperature. Our data reveal that the GFP signal is almost negligible at room temperature, while showing a very bright signal at 29°C. Therefore, we conclude that our current protocol is adequate for investigating conditional RNAi knockdown or ectopic expression of the genes in interest.

Images of *nub*>mCD8:GFP (green) and *ubi*> α -Tubulin:mCherry at 29°C or 22°C have been taken under the identical conditions.

(5) Given the potential for off-target effects, it is also imperative to demonstrate reproducibility with alternate RNAi lines, e.g. Patronin RNAi #108927 used in this study has 1 predicted off-target https://shop.vbc.ac.at/vdrc_store/108927.html.

We have tested all available Patronin RNAi stocks from Bloomington and Vienna and found that they exhibit similar phenotypes as shown below. As these RNAi stocks target different positions of the Patronin gene, we conclude that the observed Patronin RNAi phenotypes are not due to off-target effects. Using the strongest phenotypes from the RNAi stocks, we conducted a quantitative analysis. Our revised data indicate that knockdown in both the dorsal and ventral layers (*nub*>*patronin*^{RNAi}) significantly reduces the number of mitotic events, but this effect is not observed with the dorsal layer-specific RNAi (*ap*>*patronin*^{RNAi}) (revised Fig. 7). Reduced mitotic cells in *nub*>*patronin*^{RNAi} is not derived from loss of Patronin itself, but delayed loss of cell-cell contacts via uncharacterized mechanisms. In *patronin*^{RNAi} in dorsal cells only, ventral cells are wild type, thus degeneration of ECM at the basal tips of ventral protrusions occurs as usual, sufficiently regulating loss of cell-cell contacts.

Patronin-RNAi #1
V27654

Patronin-RNAi #2
V108927

Patronin-RNAi #3
BI36659

Adult wings of *nub>patronin^{RNAi}* in both dorsal and ventral epithia. Scale bars: 250 μ m.

(6) The authors show that destabilising MTs in the dorsal compartment, by expressing *Kat60*, non-autonomously disrupts mitoses in the ventral compartment. This is proposed to be indicative of cell-cell contacts being required for mitoses. Although it is possible that the disassembly process itself is required to initiate mitosis, as the IPAN is never formed in this condition, the data are not sufficient to support this conclusion. To confirm such non-autonomous roles of the IPAN, analysis should also be done following stabilisation of the microtubules specifically in the dorsal compartment using *ap-Gal4>hTau*.

As suggested by the referee, we tested overexpression of hTau only in the dorsal layer (*ap>hTau*). We found that there were no significant differences in mitoses between the control and *ap >hTau* (Fig. 6B). Although hTau overexpression seems to lead to more robust MT protrusions in dorsal cells (Fig. EV3B, D, E), our proposed model suggests that uncharacterized mechanisms regulate loss of cell-cell contacts through degeneration of ECM at the basal tips of protrusions. Since degeneration of ECM only in a single layer of epithelial cells seems to be sufficient for releasing cell-cell contacts, we interpret the data to explain why ectopic expression of hTau in two-layers or one-layer shows different mitotic phenotypes. Our proposed model to explain IPAN dynamics-mediated mechanisms are summarized in Fig. 9.

(7) The authors conclude that knockdown of Patronin leads to incomplete loss of microtubules, but again the efficiency of the RNAi for knockdown is not demonstrated. The authors need to determine whether Patronin mRNA and/or protein is depleted by qPCR or antibody staining and confirm the phenotype with an alternate Patronin RNAi line.

To validate *patronin* RNAi phenotypes, we compared the GFP-tagged Patronin expression in our live imaging protocol. We confirmed that Patronin:GFP protein was significantly reduced by RNAi knockdown (Fig. 7A, Fig. EV4A).

(8) While quantitative studies of mitoses and protrusions are referred to as "significant", this is not always obvious - e.g. from needing to compare α -spectrin RNAi which appears in Figure 4A, to the control which appears in Figure EV4D. It would be helpful to indicate statistical significance on the graphs.

In the revised manuscript, we focus on coordinated mitotic events between the two-layered epithelia. Since α -Spectrin itself affects mitosis only in a very limited manner, data related to α -Spectrin are not included. We appreciate the referee's comments.

Minor comments

(1) It would be helpful to clarify in the text what LifeAct marks, i.e. actin microfilaments

We have revised the text to indicate that LifeAct labels actin microfilaments.

(2) Although the authors show that cell protrusions include both microtubule and actin components, including lateral connections mediated by filopodia-like structures, the latter investigations focus on tubulin alone, which should be justified in the text.

In our revised manuscript, our new data reveal that lateral filopodia-like structures are actively involved in networking between vertical protrusions, contributing to the bundling of MT protrusions. Our new findings further support that bundled protrusions form either higher-order protrusions for structural support, or disassemble for coordinated mitoses. Our proposed model is summarized in Fig. 9. Overall, these observations sufficiently provide new mechanisms as well as questions to be addressed in the future.

*(3) The images in Figure 3 and EV2 are duplicated - the *cnn:RFP* panels alone should be shown in the main figure for clarity.*

This has been fixed (revised Fig. 4A).

(4) In the main text, clarify the rationale for switching to the 29 degree temperature (i.e. enabling transgene expression, overexpression in condition I and knockdown in condition II)

Rationale for temperature shift conditions are as follows and edited in the methods accordingly.

Condition I: White pupae were collected at room temperature, then shifted to 29°C until pupae were aged to 10.5h APF (equivalent to 13.5h APF at 25°C), ensuring pupal specific gene expression

Condition II: Fly vials were shifted to 29°C 16 hours prior to collecting white pupae, which in turn were kept at 29°C until their imaging commenced at 10.5h APF, ensuring pupal-specific RNAi phenotypes.

(5) Although I appreciate that different stages are written in the legends, it would be useful to have the relevant descriptions overwritten on the video EV4 alongside the arrows at 27 seconds, 31 seconds, etc.

This has been added (revised Movie EV7)

(6) There is an incorrect label on Figure 6A - the graph shows α -spectrin rather than shot RNAi?

This does not apply to the revised manuscript.

(7) The size of cells images in Figure 6D is not consistent with apical relaxation - the cells in Stage II appear the same or smaller than stage I.

This does not apply to the revised manuscript.

(8) Known/expected roles of Sqh in apical relaxation should be clarified in the text.

This does not apply to the revised manuscript.

Dear Osamu,

Thank you for submitting a revised version of your manuscript. Your study has now been seen by all original referees, who find that their previous concerns have been addressed and now recommend acceptance of the manuscript.

There now remain only a few editorial points that have to be addressed before I can extend acceptance of the manuscript:

1. Please check that the funding information is correct and identical both in the manuscript and our online system. The following grant number information appears to differ: 308045 in the manuscript and 272280 in our online system.
2. CRediT has replaced the traditional author contributions section because it offers a systematic, machine-readable author contributions format that allows for more effective research assessment. Please remove the Authors Contributions from the manuscript and use the free text boxes beneath each contributing author's name in our online submission system to add specific details on the author's contribution. More information is available in our guide to authors.
3. Please rename "Conflict of interest" section into "Disclosure and competing interests statement" (further info: <https://www.embopress.org/page/journal/14602075/authorguide#conflictsofinterest>).
4. The figure panels for Fig. 9A-D are not mentioned in the manuscript text.
5. Please remove movie legends from the manuscript file and zip each legend together with the corresponding movie file. More information is available here: <https://www.embopress.org/page/journal/14602075/authorguide#expandedview>
6. Our data editors have flagged the following issues in figure legends that need correcting:
- please define the nature of the replicates in the legends of figures 5d; 6a', c"; 7c", d", e"; EV4b.
7. Papers published in The EMBO Journal are accompanied online by a 'Synopsis' to enhance discoverability of the manuscript. It consists of A) a short (1-2 sentences) summary of the findings and their significance, B) 3-4 bullet points highlighting key results and C) a synopsis image that is 550x300-600 pixels large (width x height, jpeg or png format). You can either show a model or key data in the synopsis image. Please note that the image size is rather small and that text needs to be readable at the final size. Please send us this information together with the revised manuscript.

With best wishes,

Ieva

Ieva Gailite, PhD
Senior Scientific Editor
The EMBO Journal
Meyerohofstrasse 1
D-69117 Heidelberg
Tel: +4962218891309
i.gailite@embojournal.org

We realize that it is difficult to revise to a specific deadline. In the interest of protecting the conceptual advance provided by the work, we recommend a revision within 3 months (12th Mar 2024). Please discuss the revision progress ahead of this time with the editor if you require more time to complete the revisions.

Referee #1:

The authors have restructured the manuscript in this revised version and have focussed on the very interesting aspect of the link between the disassembly of the IPAN and timing and coordination of mitotic divisions, and they removed the alpha-Spectrin

data that previously probably distracted from this main message. The data, the imaging of this complex structure, are very high quality and beautiful and do support the conclusions drawn.

This revised manuscript, with much additional data and a clear focus now is very much improved. The authors have addressed the concerns of all three reviewers well, and I am happy with the revised manuscript.

Referee #2:

Tran et al. presented the discovery of a morphogenetic role of the microtubule-based membrane protrusion network, named the Interplanar Amida Network (IPAN), that dynamically connects the dorsal and ventral epithelium of the emerging *Drosophila* wing. The authors also showed how IPAN organization and disassembly are linked to switching microtubule organization from ncMTOC to centrosomal MTOC and coordination of cell division across different parts of the developing tissue. The authors have addressed all reviewers' concerns and suggestions, and the quality of the revised manuscript has significantly improved with new illustrations and experimental results.

I have only a minor comment about the Abstract. The following sentence appears out of context, especially without any prior mention of the non-centrosomal origin of IPAN.

"Our findings further reveal that microtubule organization switches from non-centrosomal to centrosomal microtubule organizing centers (MTOCs)."

Also, mentioning that the IPAN disassembly is linked to the switching of ncMTOC to cMTOC concomitantly with the G2/M transition of cells might provide an overall picture of how IPAN might coordinate cell division across the entire tissue.

Referee #3:

The study by Tran et al employs impressive and challenging in vivo live imaging techniques to investigate the mechanisms mediating contacts between opposed epithelial cellular layers during *Drosophila* pupal wing development. They characterise a unique dynamic cellular structure termed by the authors as Interplanar Amida Network (IPAN), and show that both the assembly and disassembly of IPAN are required to initiate mitosis in the epithelia. These novel and significant observations into 3D tissue morphogenesis would be of broad interest to developmental biologists.

In the revised manuscript, the authors expand their study to further investigate the dynamics of formation and dissociation of the IPAN, and its contribution to the regulation of cell division during pupal wing development. The authors provide interesting additional insights into the involvement of both cellular layers, and the interepithelial space including ECM, to generate and maintain inter-layer contacts. The authors omitted data relating to cell shape which has helped to streamline the manuscript, and the presented data support the conclusions.

The manuscript has been greatly improved following review and the previous concerns have been sufficiently addressed. I would like to congratulate the authors on their interesting and relevant manuscript.

The authors addressed the minor editorial issues.

Dear Osamu,

Thank you for addressing the final issues. I am now pleased to inform you that your manuscript has been accepted for publication.

Before we forward your manuscript to our publishers, I would like to propose a few minor changes in the article title, abstract and synopsis. I have also written a short blurb that will accompany the title of your manuscript in our online table of contents. Please take a look at the text below and in the attached manuscript text file and let me know if any corrections are necessary.

Title:
Programmed disassembly of a microtubule-based membrane protrusion network coordinates 3D epithelial morphogenesis in *Drosophila*

Blurb:
Microtubule-rich protrusions form contacts between two epithelial leaflets to coordinate mitosis in the developing *Drosophila* wing.

Synopsis:
During *Drosophila* pupal wing development, the dorsal and ventral epithelial layers form cytoskeleton-based protrusions that interconnect the two tissue layers. Here, in vivo live imaging of the pupal wing links the dynamics of these protrusions, named here Interplanar Amida Network (IPAN), to coordination of cell divisions across the two epithelial leaflets.

- Basal microtubule-based protrusions of the IPAN sustain cell-cell contacts between dorsal and ventral epithelia.
- IPAN disassembly during M2/S phase transition releases cell-cell contacts.
- Loss of cell-cell contact affects mitosis in both the dorsal and ventral epithelia in a coordinated manner.

If you have any questions, please do not hesitate to contact the Editorial Office. Thank you for this contribution to The EMBO Journal and congratulations on a successful publication!

Best wishes,

leva

leva Gailite, PhD
Senior Scientific Editor
The EMBO Journal
Meyerhofstrasse 1
D-69117 Heidelberg
Tel: +4962218891309
i.gailite@embojournal.org
